# Effectiveness of inactivated COVID-19 vaccines among older adults in Shanghai: retrospective cohort study

Zhuoying Huang[1,8], Shuangfei Xu[2,8], Jiechen Liu[1], Linlin Wu[1], Jing Qiu[1], Nan Wang[1], Jia Ren[1], Zhi Li[1], Xiang Guo[1], Fangfang Tao[3], Jian Chen[3], Donglei Lu[4], Yuheng Wang[5], Juan Li[1], Xiaodong Sun[1] ✉ & Weibing Wang [2,6,7] ✉

We conducted a matched retrospective cohort study of two cohorts to estimate inactivated vaccine effectiveness (VE) and its comparative effectiveness of booster dose among older people in Shanghai. Cohort 1 consisted of a vaccinated group (≥1 dose) and an unvaccinated group (3,317,475 pairs), and cohort 2 consisted of a booster vaccinated group and a fully vaccinated group (2,084,721 pairs). The Kaplan–Meier method and Cox regression models were used to estimate risk and hazard ratios (HRs) study outcomes. For cohort 1, the overall estimated VEs of ≥1 dose of inactivated vaccine against SARS-CoV-2 infection, severe/critical Covid-19, and Covid-19 related death were 24.7% (95% CI 23.7%–25.7%), 86.6% (83.1%–89.4%), and 93.2% (88.0%–96.1%), respectively. Subset analysis showed that the booster vaccination provided greatest protection. For cohort 2, compared with full vaccination, relative VEs of booster dose against corresponding outcome were 16.3% (14.4%–17.9%), 60.5% (37.8% −74.9%), and 81.7% (17.5%−95.9%). Here we show, although under the scenario of persistent dynamic zero-Covid policy and non-pharmaceutical interventions, promoting high uptake of the full vaccination series and booster dose among older adults is critically important. Timely vaccination with the booster dose provided effective protection against Covid-19 outcomes.

Through January 10, 2023, there have been over 660 million confirmed cases of coronavirus disease 2019 (Covid-19), and over 6.6 million COVID-19-related deaths reported to the World Health Organization (WHO)[1]. Over 170 SARS-CoV-2 vaccines are in various stages of clinical development; thirteen vaccines have been emergency-use listed by WHO and nearly 13 billion doses of COVID-19 vaccines have been administered[2,3]. Most published vaccine effectiveness (VE) evidence is

about mRNA vaccines[4–8]. In mainland China, 15 Covid-19 vaccines have been approved, either conditionally or for emergency use. Inactivated vaccines (Sinovac's CoronaVac, Sinopharm's BIBP-CorV, and Sinopharm's WIBP-CorV) are the most extensively used vaccines in China and are widely used globally. Randomized controlled trials and real-world studies have shown that inactivated Covid-19 vaccines provide weak protection against SARS-CoV-2 infection but promising results

[1]Institute of Immunization, Shanghai Municipal Center of Disease Control and Prevention, Shanghai 200336, China. [2]Shanghai Institute of Infectious Disease and Biosecurity, School of Public Health, Fudan University, Shanghai 200032, China. [3]Institute of Infectious Diseases, Shanghai Municipal Center of Disease Control and Prevention, Shanghai 200336, China. [4]Division of Health Risk Factors Monitoring and Control, Shanghai Municipal Center of Disease Control and Prevention, Shanghai 200336, China. [5]Division of Chronic Non-communicable Diseases and Injury, Shanghai Municipal Center of Disease Control and Prevention, Shanghai 200336, China. [6]Key Laboratory of Public Health Safety of Ministry of Education, Fudan University, Shanghai 200032, China. [7]Key Laboratory of Health Technology Assessment, Fudan University, Shanghai 200032, China. [8]These authors contributed equally: Zhuoying Huang, Shuangfei Xu. ✉e-mail: sunxiaodong@scdc.sh.cn; wwb@fudan.edu.cn

against severe or fatal illness, hospitalization, and death in adults— including during times of Omicron variant predominance[9–12]. Two doses of CoronaVac reduced by 79.3% severe or fatal Covid-19 and by 84.3% Covid-19 death among those aged 60–69 years, and by 58.2% severe or fatal Covid-19 and by 63.0% Covid-19 death among those aged 80 years or older[11]. Eight to 59 days after receiving a booster dose, CoronaVac reduced 8.6% of remaining symptomatic disease and 73.6% of remaining severe Covid-19[13]. However, there is still a paucity of evidence on inactivated vaccine effectiveness and comparative effectiveness of inactivated vaccine booster doses, especially among older individuals.

In March 2021, Shanghai started vaccinating individuals 60–75 years old, and in May 2021, individuals 76 years and older. Booster doses were authorized in November 2021 for adults who received their primary series at least 6 months earlier. The recommended interval between doses of primary series inactivated vaccine was 21–56 days[14,15]. Vaccination campaigns were voluntary and with informed consent. Our previous work showed the willingness to vaccinate the older in their family was around 40% in Shanghai[16]. In Shanghai, as of May 13, 2022, fewer than 70% of 60–79-year-olds completed a primary series, with 40% receiving a booster dose. Among adults over 80 years of age, 15% and 10% completed primary series and booster vaccination, respectively[12]. Between March 2022 and May 2022, an outbreak of the Omicron BA.2 variant occurred in Shanghai that led to 1376 severe or critical illnesses and 544 deaths among adults 60 years or older, accounting for over 90% of all severe outcomes in the outbreak. The citywide vaccination campaign was suspended during the outbreak[12]. The epidemic prevention and control strategy of Covid-19 changed from a targeted approach (March 2021 to March 2022) to strict non-

pharmaceutical interventions (NPIs), such as citywide home quarantine, massive nucleic acid amplification testing/rapid antigen testing (NAAT/RAT), and centralized quarantine of close contacts (March 2022 to May 2022), to regular, periodic, population-wide routine NAAT (May 2022 to Dec 7, 2022). Evidence regarding the effectiveness of inactivated vaccine and comparative effectiveness of a booster is needed to inform vaccination policies and guide subsequent response.

We obtained data reported to the integrated data repositories of Shanghai Center for Disease Control and Prevention between March 25, 2021 and July 16, 2022 to evaluate effectiveness of inactivated Covid-19 vaccines available in China against three outcomes: documented SARS-CoV-2 infection, severe/critical Covid-19, and Covid-19 related death. We used this observational dataset to conduct a retrospective, matched cohort study to evaluate the effectiveness of ≥1 dose of inactivated vaccine and the comparative effectiveness of a booster dose in a community-dwelling population aged 60 years and older, and in subpopulations defined by age, sex, and coexisting medical conditions.

## Results
### Study population

Figure 1 shows the vaccination status and cumulative documented SARS-CoV-2 infections among adults aged 60 years and above in Shanghai, 25 March 2021 through 16 July 2022. Since 12 March 2022, a rapid spread of SARS-CoV-2 occurred in Shanghai covering nearly 96% of documented cases in this study. By the start of the study period, 5,738,317 individuals were assessed for eligibility; 5,438,759 individuals 60 years or older who received at least one dose of an inactivated vaccine were included in the vaccinated group and matched to

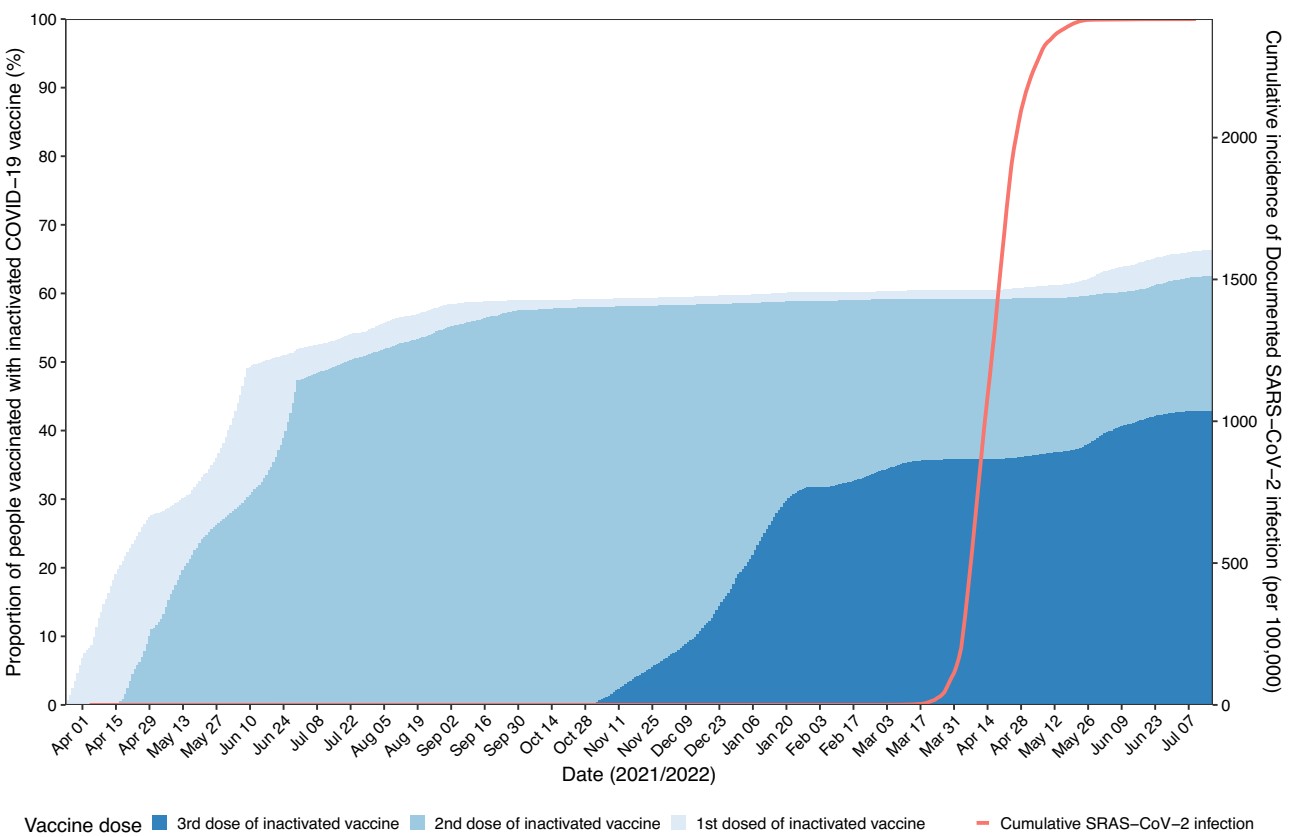

**Fig. 1 | Vaccination status and cumulative documented SARS-CoV-2 infections among adults aged ≥60 years in Shanghai (25 March 2021 to 16 July 2022).** Dusty blue, light blue, and royalblue represent the proportion of people vaccinated with first dose, second dose, and third dose of inactivated Covid-19 vaccine, respectively. Red line shows the cumulative incidence of documented SARS-CoV-2 infections.

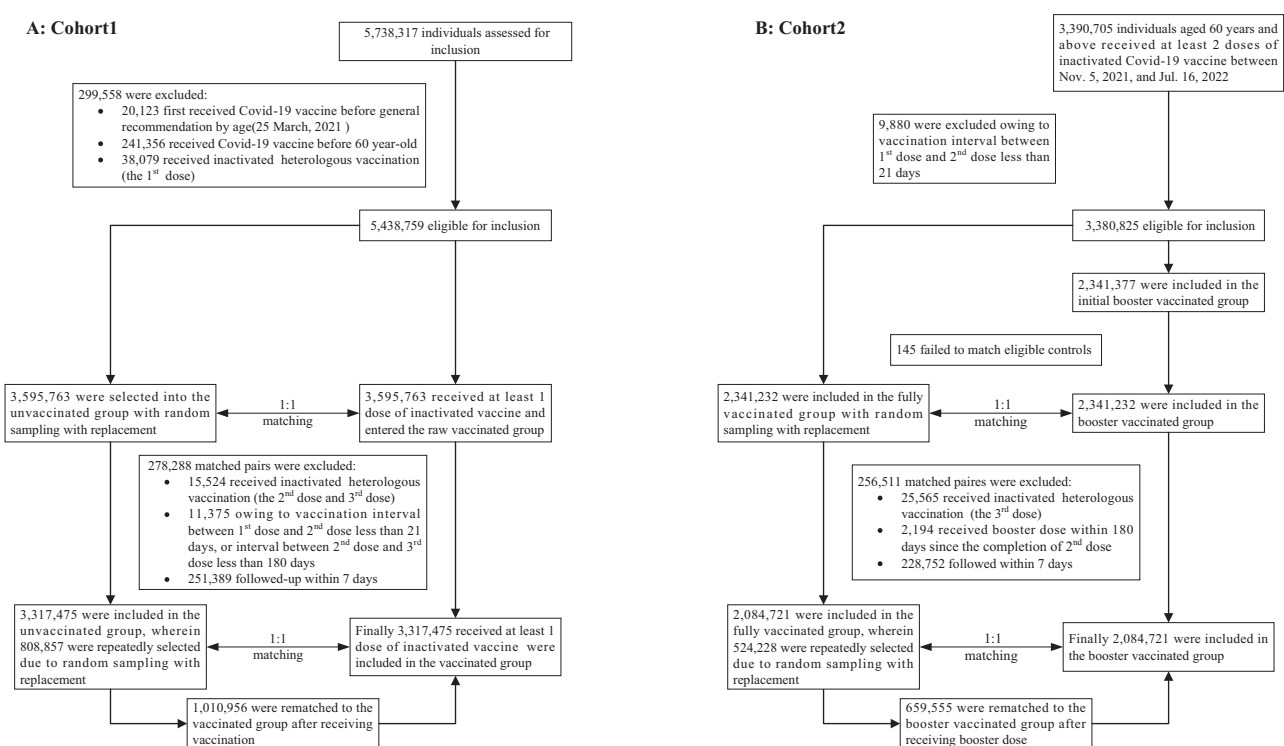

**Fig. 2 | Cohort enrollment process.** Cohort selection for the analysis of effectiveness of vaccination with inactivated vaccine (**A:** Cohort 1) and booster vaccination with inactivated vaccine (**B:** Cohort 2).

unvaccinated controls in cohort 1. Between November 5, 2021, and July 16, 2022, 3,390,705 individuals 60 years or older had received at least two doses of inactivated vaccine; among these individuals, 2,341,232 received a homologous third (booster) dose and were included in the booster vaccinated group and matched to fully vaccinated controls in cohort 2. The selection process is shown in Fig. 2. In cohort 1, 30.5% of unvaccinated controls and their matched pairs were censored when the controls received a dose of vaccine. In cohort 2, 31.6% of fully vaccinated controls and their matched paired were censored when the control received a booster dose.

Baseline characteristics of the matched pairs are shown in Table 1. In cohort 1, the median age was 67 years (IQR 63–71), and 2,253,375 (67.9%) were between 60 and 69 years; 1,640,022 (49.4%) were female; 1,306,957 (39.4%) had ≥1 chronic disease. In the vaccinated group, 570,830 (17.2%), 1,587,415 (47.8%), and 1,159,230 (34.9%) completed partial vaccination, full vaccination, and booster vaccination, respectively. In cohort 2, the median age was 67 years (IQR 64–71), and 1,410,798 (67.7%) were between 60 and 69 years; 1,057,073 (50.7%) were female; 785,245 (37.7%) had ≥1 chronic disease; 2,069,237 (99.3%) received their second dose between 21 and 56 days since the receipt of first dose; and 1,720,566 (82.5%) received a booster dose 181–270 days after receipt of their second dose.

Baseline demographic characteristics of cohort 1 and cohort 2 in sensitivity analysis, where controls were selected with sampling without replacement, are shown in Table S1.

## At least one-, at least two-, or three-dose vaccine effectiveness versus no vaccination

During a median [IQR] follow-up time of 281.0 [46.0, 409.0] days for cohort 1, after 8 or more days since receipt of vaccine or since becoming a matched control, 85,028 SARS-CoV-2 infections (382.4 per million person-weeks [PWs] of follow-up) were documented, with 48,447 in the unvaccinated group and 36,581 in the vaccinated group; 682 infections progressed to severe/critical Covid-19, 601 in the unvaccinated group and 81 in the vaccinated

group; and 203 were Covid-19 related deaths, 190 in the unvaccinated group and 13 in the vaccinated group. Figure 3 shows cumulative incidence curves for the three primary outcomes. Table 2 shows the estimated VEs for the outcomes in the whole study population and in subpopulations defined by age group, sex, and chronic disease status. Overall, the estimated VE of ≥1 dose of inactivated vaccine against SARS-CoV-2 infection was 24.7% (95% CI 23.7–25.7); VE against severe/critical Covid-19 was 86.6% (95% CI 83.1–89.4); and VE against Covid-19 related death was 93.2% (95% CI 88.0–96.1). Table 3 shows the estimated VEs of at least two-dose vaccination, and three-dose vaccination versus no vaccination. We found the booster vaccination provided greatest protection against the main outcomes. Table S2 showed the VEs of sensitivity analyses. Table S3 showed the VEs by vaccine brand.

## Booster dose effectiveness versus full vaccination

The median follow-up time was 76 [IQR 30, 178] days for the booster vaccinated group and 75.0 [IQR 30.0, 177.0] for the fully vaccinated group in cohort 2. A total of 19,470 infections (648.7 per million PWs) was documented in the booster vaccinated group 8 days or more after receipt of the booster dose. Twenty-six of these infections progressed to severe/critical Covid-19, and two died from Covid-19. A total of 23,193 infections (774.9 per million PWs) occurred in the fully vaccinated group. Among these infections, 66 progressed to severe/critical Covid-19, and 11 progressed to Covid-19-related death. Figure 4 shows cumulative incidence curves for study outcomes. Table 4 shows the estimated VEs for the study outcomes for the whole study population and for subpopulations defined by age group, sex, chronic disease status, time interval between dose 1 and dose 2, and time since receipt of the second dose. Compared with full vaccination, relative VEs of booster dose were 16.3% (95% CI 14.4–17.9) against SARS-CoV-2 infection, 60.5% (95% CI 37.8–74.9) against severe/critical Covid-19, and 81.7% (95% CI 17.5–95.9) against Covid-19 related death.

**Table 1 | Baseline demographic characteristics of cohort 1 and cohort 2**

| | Cohort 1 Unvaccinated group (N = 3,317,475) | Vaccinated group (N = 3,317,475) | SMD | Cohort 2 Fully vaccinated group (N = 2,084,721) | Booster vaccinated group (N = 2,084,721) | SMD |
|---|---|---|---|---|---|---|
| Median age [IQR]—years old | 67.0 [63.0, 71.0] | 67.0 [63.0, 71.0] | <0.01 | 67.0 [64.0, 71.0] | 67.0 [64.0, 71.0] | <0.01 |
| **Age group—no. (%)** | | | | | | |
| 60–69 years old | 2,253,375 (67.9) | 2,253,375 (67.9) | <0.01 | 1,410,798 (67.7) | 1,410,798 (67.7) | <0.01 |
| 70–79 years old | 876,363 (26.4) | 876,363 (26.4) | | 598,241 (28.7) | 598,241 (28.7) | |
| 80–89 years old | 175,716 (5.3) | 175,716 (5.3) | | 73,230 (3.5) | 73,230 (3.5) | |
| 90 years old and above | 12,021 (0.4) | 12,021 (0.4) | | 2,452 (0.1) | 2,452 (0.1) | |
| **Sex—no. (%)** | | | | | | |
| Male | 1,677,453 (50.6) | 1,677,453 (50.6) | <0.01 | 1,027,648 (49.3) | 1,027,648 (49.3) | <0.01 |
| Female | 1,640,022 (49.4) | 1,640,022 (49.4) | | 1,057,073 (50.7) | 1,057,073 (50.7) | |
| **Chronic disease—no. (%)** | | | | | | |
| 0 | 2,010,518 (60.6) | 2,010,518 (60.6) | <0.01 | 1,299,476 (62.3) | 1,299,476 (62.3) | <0.01 |
| ≥1 | 1,306,957 (39.4) | 1,306,957 (39.4) | | 785,245 (37.7) | 785,245 (37.7) | |
| Cancer | 253,113 (7.6) | 14,4130 (4.3) | 0.14 | 81,970 (3.9) | 75,429 (3.6) | 0.02 |
| Hypertension | 1,034,212 (31.2) | 1,098,747 (33.1) | 0.04 | 663,094 (31.8) | 664,922 (31.9) | <0.01 |
| Diabetes | 376,072 (11.3) | 338,045 (10.2) | 0.04 | 196,338 (9.4) | 201,386 (9.7) | 0.01 |
| **Vaccination status** | | | | | | |
| Unvaccinated | 3,317,475 (100.0) | – | – | – | – | – |
| Partial vaccination | – | 570,830 (17.2) | | – | – | |
| Full vaccination | – | 1,587,415 (47.9) | | 2,084,721 (100.0) | – | |
| Booster vaccination | – | 1,159,230 (34.9) | | – | 2,084,721 (100.0) | |
| **Vaccine brand** | | | | | | |
| Unvaccinated | 3,317,475 (100.0) | – | – | – | – | – |
| One dose Sinovac-CorV | – | 299,708 (9.0) | | – | – | |
| One dose BIBP-CorV | – | 260,155 (7.8) | | – | – | |
| One dose WIBP-CorV | – | 10,620 (0.3) | | – | – | |
| One dose other inactive vaccine | – | 347 (0.0) | | – | – | |
| Two-dose Sinovac-CorV | – | 866,900 (26.1) | | 1,178,817 (56.5) | – | |
| Two-dose BIBP-CorV | – | 675,739 (20.4) | | 853,789 (41.0) | – | |
| Two-dose WIBP-CorV | – | 14,333 (0.4) | | 17,981 (0.9) | – | |
| Mixed two doses inactive vaccine | – | 30,443 (0.9) | | 34,134 (1.6) | – | |
| Three-dose Sinovac | – | 634,001 (19.1) | | – | 1,142,717 (54.8) | |
| Three-dose BIBP-CorV | – | 480,233 (14.5) | | – | 857,193 (41.1) | |
| Three-dose WIBP-CorV | – | 11,049 (0.3) | | – | 26,217 (1.3) | |
| Mixed three doses inactive vaccine | – | 33,947 (1.0) | | – | 58,594 (2.8) | |
| **Time interval between 1st dose and 2nd dose** | | | | | | |
| Median [IQR]—days | – | – | | 23.0 [21.0, 27.0] | 23.0 [21.0, 27.0] | <0.01 |
| 21–56 days | – | – | | 2,069,237 (99.3) | 2,069,237 (99.3) | <0.01 |
| 57–180 days | – | – | | 15,074 (0.7) | 15,074 (0.7) | |
| 181 days and above | – | – | | 410 (0.0) | 410 (0.0) | |
| **Time interval since completion of 2nd dose** | | | | | | |
| Median [IQR]—days | – | – | | 218.0 [197.0, 252.0] | 218.0 [198.0, 251.0] | <0.01 |
| 181–270 days | – | – | | 1,720,566 (82.5) | 1,720,566 (82.5) | <0.01 |
| 271–360 | – | – | | 250,800 (12.0) | 250,800 (12.0) | |
| 361 days and above | – | – | | 113,355 (5.4) | 113,355 (5.4) | |

*SMD* standardized mean difference.

## Discussion

Among vaccinated individuals, adjusted VEs of at least one dose were 24.7% against SARS-CoV-2 infection, 86.6% against severe/critical Covid-19, and 93.2% against Covid-19-related death. VE estimates among individuals with coexisting medical conditions were slightly higher. Compared with receipt of two doses of inactivated vaccine at least 181 days prior, administering a third homologous dose was associated with a 16.3% reduction in the incidence of SARS-CoV-2 infection, a 60.5% reduction in the incidence of severe/critical illness, and a 81.7% reduction in the incidence of Covid-19 related death.

Performance of one or more doses of inactivated vaccine was similar in each subpopulation analysis, notably effective at preventing severe/critical Covid-19 and Covid-19 related death. We observed a severe disease rate of 0.8% and a case fatality rate of less than 0.2% in cohort 1 during the study period—much lower rates than those observed in other countries and regions[11,17,18]. This

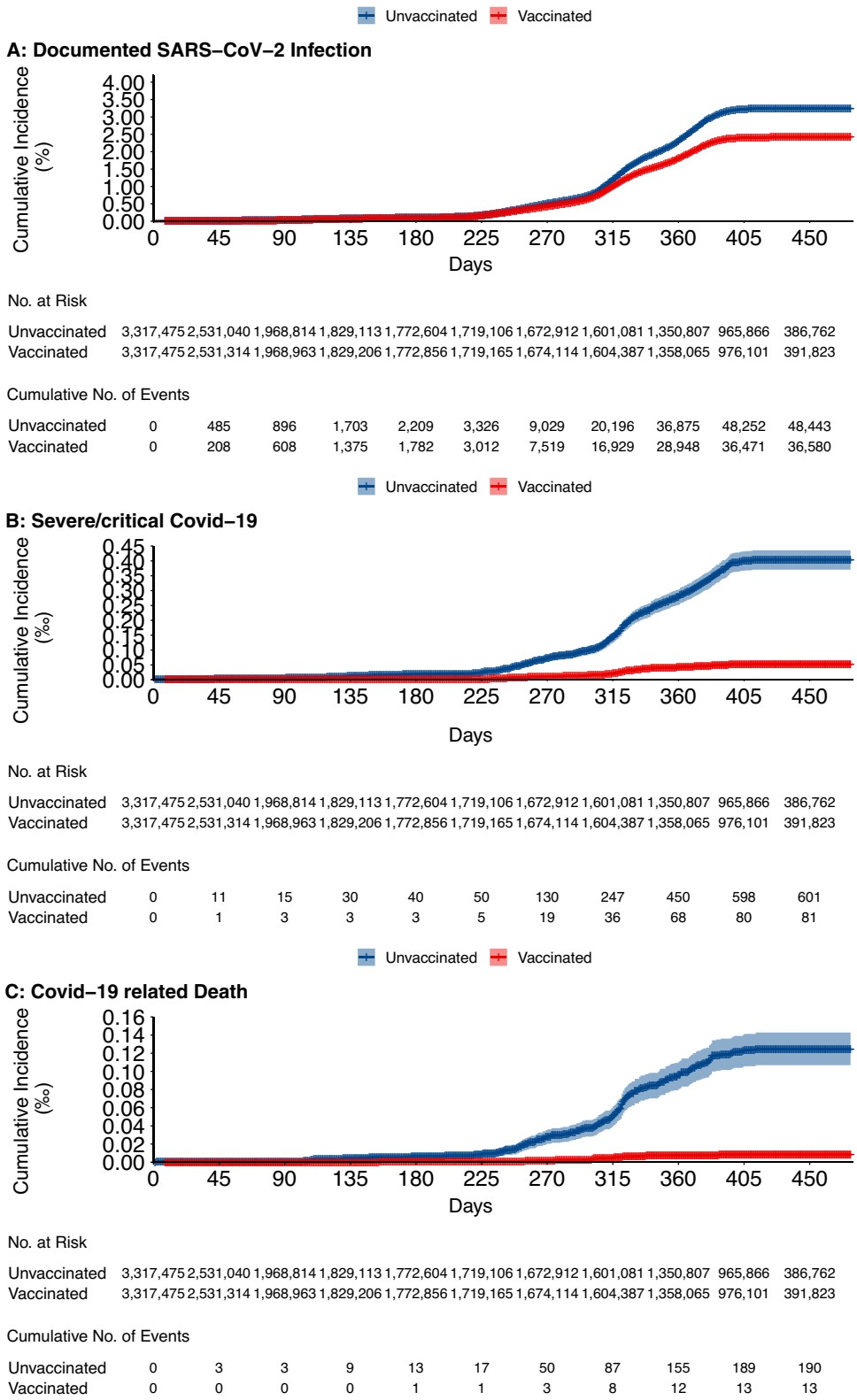

**Fig. 3 | Cumulative incidence of three outcomes in cohort 1. A** SARS-CoV-2 infection, **B** Severe/critical Covid-19, and **C** Covid-19 related death.

phenomenon is perhaps due to the lower pathogenicity of Omicron variant as seen in mice and hamsters models and in human data[19–21] and, consistent with previous studies[22], due to prompt medical assistance from surrounding provinces and cities and a persistent dynamic zero-Covid policy based on non-pharmaceutical interventions (NPIs). In addition, massive screening in strict lockdown period (Mar 30, 2022 to May 31, 2022) achieved early discovery,

diagnosis, and treatment for infection, and cases had a milder clinical severity. In Table 3, which shows results when three subsets were selected from cohort 1, we found that vaccine protection was better provided by three-dose regimens than two- or one-dose. Overall booster VE against infection stabilized at 40% after six months. Full vaccination contributed a 67.1% reduction in the incidence of severe/critical Covid-19 and an 81.2% reduction in Covid-19

**Table 2 | Adjusted vaccine effectiveness of inactivated Covid-19 vaccine against Covid-19 outcomes**

| | Unvaccinated (N = 3,317,475) | | Vaccinated (N = 3,317,475) | | Risk difference per million PWs (95% CI) | VE (95% CI) |
|---|---|---|---|---|---|---|
| | Events | Risk per million PWs | Events | Risk per million PWs | | |
| **Outcome 1: Documented SARS-CoV-2 infection** | | | | | | |
| Overall | 48,447 | 436.1 | 36,581 | 328.8 | 107.3 (106.8, 107.8) | 24.7 (23.7, 25.7) |
| Age group—no. | | | | | | |
| 60–69 years old | 32,458 | 437.2 | 25,124 | 337.9 | 99.3 (98.7, 99.9) | 22.8 (21.5, 24.0) |
| 70–79 years old | 13,332 | 423.7 | 9849 | 312.5 | 111.2 (110.2, 112.2) | 26.4 (24.5, 28.3) |
| 80–89 years old | 2501 | 481.5 | 1532 | 294.2 | 187.3 (183.2, 191.4) | 38.9 (34.9, 42.7) |
| 90 years old and above | 156 | 794.8 | 76 | 385.3 | 409.6 (371.4, 447.6) | 52.1 (37.0, 63.6) |
| Sex—no. (%) | | | | | | |
| Male | 23,800 | 427 | 18,288 | 327.6 | 99.4 (98.7, 100.1) | 23.4 (21.9, 24.9) |
| Female | 24,647 | 445.2 | 18,293 | 329.9 | 115.3 (114.5, 116.1) | 25.9 (24.5, 27.4) |
| Chronic disease—no. | | | | | | |
| 0 | 27,738 | 433.9 | 22,001 | 343.7 | 89.6 (89.0, 90.1) | 20.9 (19.5, 22.3) |
| ≥1 | 20,709 | 439.1 | 14,580 | 308.5 | 129.4 (128.5, 130.4) | 29.8 (28.2, 31.3) |
| **Outcome 2: Severe/critical Covid-19** | | | | | | |
| Overall | 601 | 5.4 | 81 | 0.7 | 4.7 (4.4, 5.0) | 86.6 (83.1, 89.4) |
| Age group—no. (%) | | | | | | |
| 60–69 years old | 250 | 3.4 | 20 | 0.3 | 3.1 (2.8, 3.4) | 91.8 (87.1, 94.8) |
| 70–79 years old | 201 | 6.4 | 39 | 1.2 | 5.2 (4.7, 5.6) | 80.7 (72.7, 86.3) |
| 80–89 years old | 135 | 26.0 | 18 | 3.5 | 22.5 (19.7, 25.3) | 86.7 (78.3, 91.9) |
| 90 years old and above | 15 | 76.4 | 4 | 20.3 | 56.1 (37.3, 74.1) | 73.6 (20.4, 91.2) |
| Sex—no. (%) | | | | | | |
| Male | 184 | 3.3 | 24 | 0.4 | 2.9 (2.6, 3.2) | 86.8 (79.8, 91.4) |
| Female | 417 | 7.5 | 57 | 1.0 | 6.5 (6.0, 7.0) | 86.5 (82.2, 89.8) |
| Chronic disease—no. | | | | | | |
| 0 | 269 | 4.2 | 32 | 0.5 | 3.7 (3.4, 4.0) | 88.1 (82.9, 91.8) |
| ≥1 | 332 | 7.0 | 49 | 1.0 | 6.0 (5.5, 6.5) | 85.4 (80.2, 89.2) |
| **Outcome 3: Covid-19-related death** | | | | | | |
| Overall | 190 | 1.7 | 13 | 0.1 | 1.6 (1.4, 1.8) | 93.2 (88.0, 96.1) |
| Age group—no. (%) | | | | | | |
| 60–69 years old | 64 | 0.9 | 2 | 0.0 | 1.2 (0.9, 1.4) | 96.8 (87.0, 99.2) |
| 70–79 years old | 67 | 2.1 | 7 | 0.2 | 2.0 (1.7, 2.3) | 89.5 (77.1, 95.2) |
| 80–89 years old | 52 | 10.0 | 3 | 0.6 | 1.2 (0.9, 1.4) | 94.2 (81.5, 98.2) |
| 90 years old and above | 7 | 35.7 | 1 | 5.1 | 2.0 (1.7, 2.3) | 85.8 (−15.5, 98.3) |
| Sex—no. | | | | | | |
| Male | 68 | 1.2 | 3 | 0.1 | 1.1 (0.9, 1.3) | 95.6 (86.1, 98.6) |
| Female | 122 | 2.2 | 10 | 0.2 | 2.0 (1.7, 2.3) | 91.8 (84.3, 95.7) |
| Chronic disease—no. (%) | | | | | | |
| 0 | 108 | 1.7 | 3 | 0.0 | 1.6 (1.4, 1.9) | 97.2 (91.3, 99.1) |
| ≥1 | 82 | 1.7 | 10 | 0.2 | 1.5 (1.3, 1.8) | 87.8 (76.3, 93.7) |

*95% CI* 95% confidence interval, *PW* person-weeks, *NA* not available.
Vaccine effectiveness was adjusted by presence of comorbidities of cancer, hypertension, or diabetes.

related death. Booster vaccination contributed to an 88.1% reduction of severe/critical Covid-19 and 100% reduction of death. These findings were similar to inactivated VEs reported in Brazil[13] and Hong Kong[11], and were slightly higher than RNA-based COVID-19 VEs reported in the UK (three-dose mRNA vaccine: 76.9% to 95.8% against hospitalization)[23], USA (three-dose any mRNA vaccine: 93% against hospitalization)[24], and Israel (three-dose BNT162b2: 81.6% against severe or critical disease, and 77.1% against death)[25].

In cohort 2 the estimated relative VEs of a third dose of inactivated vaccine (16.3% against infection, 60.5% against severe/critical Covid-19, and 81.7% against Covid-19-related death) were lower than those from mRNA vaccines as observed in Spain (BNT162b2: 46.2% against infection; mRNA-1273: 52.3% against infection)[7] and Qatar (BNT162b2: 49.4% against infection and 76.5% against

hospitalization; mRNA-1273: 47.3% against infection)[6]. A matched test-negative design study in Brazil estimating relative VE of a booster dose showed that protection with a heterologous booster using an mRNA vaccine was more effective and durable than that from homologous booster with CoronaVac[13]. In mainland China, however, little evidence exists on primary or booster vaccination with mRNA vaccine.

As seen in Table 3, there was an unexpected finding of "negative" VEs of full vaccination series against SARS-CoV-2 infection—a phenomenon observed in other Covid-19 vaccine studies[13,26]. When analyzing the full vaccination group by follow-up times of 8–60 days, 61–80 days, and ≥181 days, we found the first two time-varying VEs against infection were 94.1% (95% CI 91.1–96.1%) and 11.9% (4.6–18.7%), which were higher values than estimates reported in Brazil for

**Table 3 | Adjusted vaccine effectiveness against Covid-19 outcomes of at least two-dose, and three-dose of inactivated Covid-19 vaccines versus no vaccination**

| | Subset 1: at least two-dose group VE (95% CI) | | | Subset 2: three-dose group VE (95% CI) | | |
|---|---|---|---|---|---|---|
| | Documented SARS-CoV-2 Infection | Severe/critical Covid-19 | Covid-19 related death | Documented SARS-CoV-2 Infection | Severe/critical Covid-19 | Covid-19 related death |
| Overall | 25.9 (24.9, 26.9) | 87.5 (84.0, 90.2) | 94.0 (88.9, 96.7) | 47.6 (46.7, 48.6) | 94.9 (92.0, 96.7) | 99.2 (94.2, 99.9) |
| Age group | | | | | | |
| 60–69 years old | 23.9 (22.6, 25.2) | 92.5 (87.9, 95.4) | 96.8 (87.0, 99.2) | 45.2 (44.0, 46.5) | 96.0 (91.0, 98.3) | 100 (NA) |
| 70–79 years old | 27.8 (25.8, 29.7) | 82.9 (75.3, 88.2) | 90.3 (77.6, 95.8) | 50.6 (48.8, 52.3) | 95.6 (90.0, 98.1) | 100 (NA) |
| 80–89 years old | 40.4 (36.3, 44.2) | 85.9 (76.9, 91.4) | 95.9 (83.1, 99.0) | 59.9 (56.1, 63.3) | 94.2 (85.7, 97.7) | 100 (NA) |
| 90 years old and above | 53.2 (37.8, 64.8) | 71.6 (13.9, 90.7) | 85.8 (–15.5, 98.3) | 69.0 (53.3, 79.5) | 66.8 (-22.5, 91.0) | 80.2 (–69.4, 97.7) |
| Sex | | | | | | |
| Male | 24.6 (23.1, 26.0) | 86.6 (79.2, 91.3) | 95.3 (84.9, 98.5) | 48.6 (47.2, 50.0) | 92.3 (84.1, 96.2) | 97.4 (81.3, 99.6) |
| Female | 27.2 (25.7, 28.6) | 87.8 (83.6, 90.9) | 93.2 (86.2, 96.7) | 46.7 (45.3, 48.1) | 95.8 (92.5, 97.6) | 100 (NA) |
| Chronic disease | | | | | | |
| 0 | 22.2 (20.8, 23.6) | 89.3 (84.2, 92.8) | 97.1 (90.9, 99.1) | 44.3 (42.9, 45.6) | 97.1 (92.9, 98.8) | 100 (NA) |
| ≥1 | 30.8 (29.3, 32.3) | 86.0 (80.8, 89.7) | 89.7 (78.6, 95.0) | 52.0 (50.6, 53.4) | 93.1 (88.3, 95.9) | 98.1 (86.5, 99.7) |

*NA* not available.

preventing symptomatic Covid-19 (28.1%, 3.9%)[13], but we found no protection when ≥181 days (6.3% in Brazil). The difference may possibly be due to unawareness of signs/symptoms for SARS-CoV-2 infection in Shanghai and differences in local NPI intensity and transmission dynamics. These slightly negative VE results may also be related to higher contact levels between vaccinated individuals[27] or to other uncontrolled factors. We found rapid waning of protection against infection. We also noticed wide confidence intervals of VEs against severe Covid-19 outcomes in sub-age-groups (defined by 10-year bands) population analyses, which reflect the relatively large population size and the relatively small number of cases in Shanghai. In addition, there was a disparity in the number of booster dose recipients between the two cohorts (1,159,230 in cohort 1 and 2,084,721 in cohort 2), which likely arose from control-group individuals' vaccination behavior. Many unvaccinated controls in cohort 1 got vaccinated in the later part of the study period (the overall vaccination coverage in adults 60 years of age and older was 63%), and their vaccinated matches had to be censored with their vaccination status recorded as "partially vaccinated" or "fully vaccinated" even though they actually received their second and third doses. However, cohort 2 did not have an analogous situation, and was able to include almost all eligible booster dose recipients. In our study, we observed VEs among sub-populations "free of chronic disease" were slightly lower than among subpopulations with one or more chronic diseases. This finding is inconsistent with what observed in immunogenicity[28], or in some population VEs studies[29]. This finding may be because we did not fully identify subpopulations "free of chronic disease", or perhaps individuals with coexisting conditions may have reduced activity spheres and consciously decrease their social contacts, especially under the COVID-19 epidemic.

Our study has several strengths. We utilized a massive health care dataset that covers almost all adults aged 60 years and older in Shanghai. By integrating data from the vaccination system, the disease prevention and control system, and the chronic disease passive surveillance system, we were able to conduct a matched cohort study design in a large, real-world population and estimate VEs of vaccination, full vaccination, booster vaccination, and booster VEs relative to full vaccination against various Covid-19 outcomes. The matching process was performed daily, thus, the difference in exposure risk was partly reduced. Additionally, implementation of emergency citywide NAAT followed by routine NAATs every 48 h in Shanghai ensures complete SARS-CoV-2 infection identification. Finally, follow-up times

in our study were longer than previous study and allowed us to estimate VEs against various Covid-19 outcomes.

Our study has several limitations. First, the chronic disease data available for this study was collected as part of Shanghai Center for Disease Control and Prevention's passive surveillance efforts and was restricted to cancer, hypertension, and type II diabetes mellitus. Therefore, not all comorbidities were recorded and available for use in the adjustment in Cox regression analyses. As is often done in population-based studies based on record linkages, we considered those not linked to the chronic disease database as having no comorbidity, leading to some misclassification of coexisting conditions. However, matching by years of age will have reduced bias due to comorbidity status misclassification somewhat. Second, matching was performed to control for confounders that might affect the exposure to SARS-CoV-2 infection and vaccination behaviors. However, some key variables, such as occupational exposure, neighborhood or other spatial features, living environment (community or nursing home), were not available and may not have been balanced, leading to an unknown bias in an unknown direction. Third, owing to the relatively long follow-up (8–478 days in cohort 1, and 8–274 days in cohort 2), Shanghai experienced variation in control and prevention intensity against Covid-19 epidemics. It is possible that these changes biased the VE estimation by decreasing virus transmission. However, our daily matching would have reduced differential control measure exposures in the study cohorts, and relatively small number of Covid-19-related deaths in our study also resulted wide confidence interval, which influenced the result precious. Finally, our study was observational and study population to unmeasured confounding. Vaccinated persons may differ in key characteristics from unvaccinated persons, leading to bias, which may change over time.

Although our study was conducted under a scenario of persistent dynamic zero-Covid policy and non-pharmaceutical interventions, the findings have somewhat implications for the current vaccination strategy in China in the context of the current global Covid-19 epidemic: it is critically important to promote high uptake of the full vaccination series with booster dose administration among adults 60 years of age or older without SARS-CoV-2 infection history, especially in vulnerable individuals with coexisting medical conditions. Timely vaccination with the booster dose provided effective protection against Covid-19 outcomes.

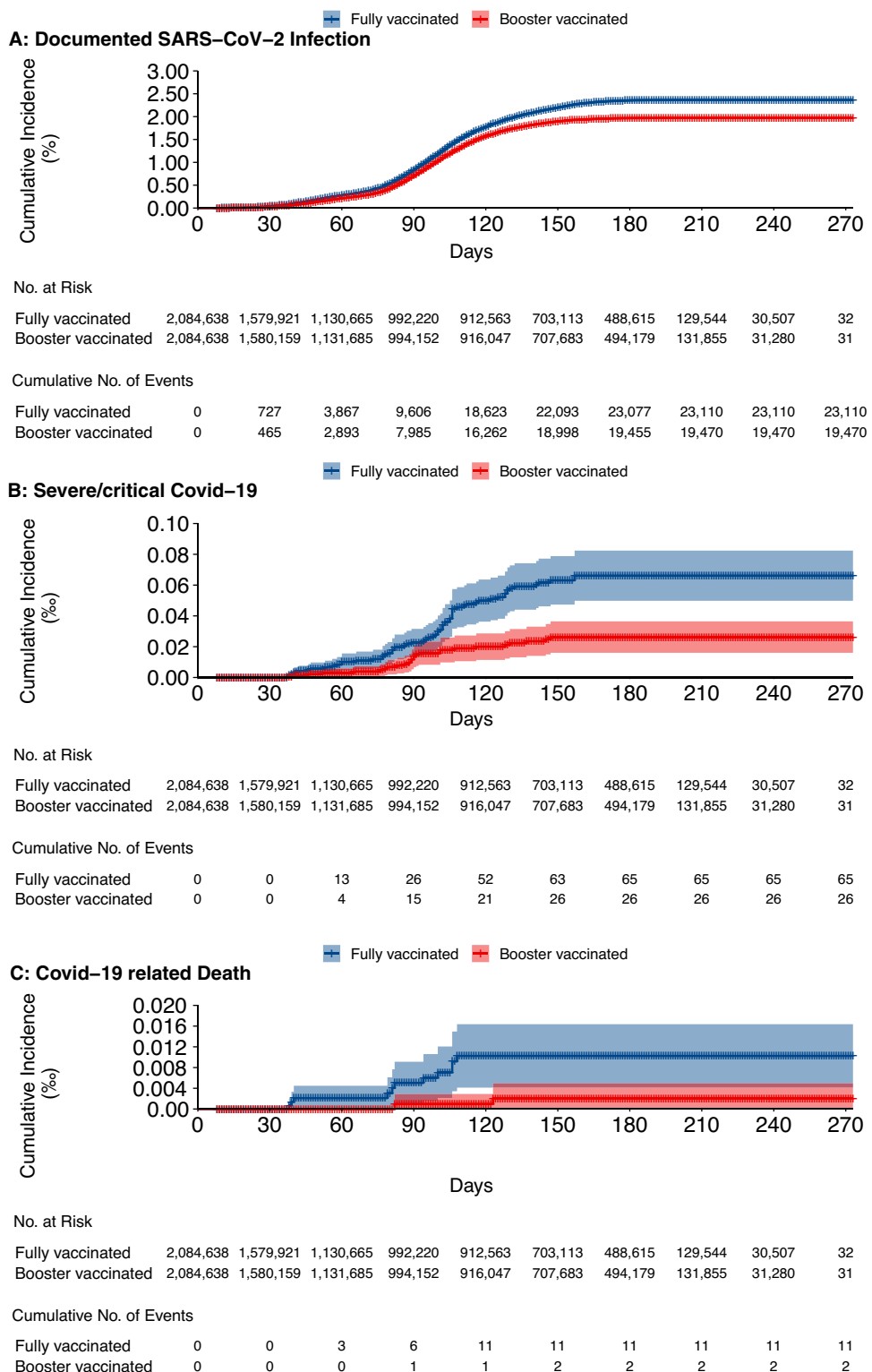

**Fig. 4 | Cumulative incidence of three outcomes in cohort 2. A** SARS-CoV-2 infection, **B** severe/critical Covid-19, and **C** Covid-19 related death.

## Methods

### Data sources

The source population were derived from individuals undergoing several rounds of citywide massive NAAT/RAT, which involved everyone living in Shanghai including citizens, foreigners, and immigrants. The study was conducted among adults aged 60 years and above residing in Shanghai between March 25, 2021 (the starting date of vaccination campaign of people over 60 years in Shanghai) and July 16, 2022 (the

ending date of the study period). Individual-level data were obtained from the Shanghai Group Immunization System and National Immunization Program Information System. Infection data were obtained from the National Notifiable Diseases Reporting System (NNDRS). Starting April 2022, several rounds of citywide nucleic acid amplification testing (NAAT) were conducted, followed by routine NAAT in which nearly all citizens in Shanghai were SARS-CoV-2 RT-PCR tested every 48 hours, regardless of presence or absence of Covid-19 symptoms.

**Table 4 | Adjusted relative vaccine effectiveness of the booster vaccination comparing to full vaccination of inactivated Covid-19 vaccine**

| | Fully vaccinated group (*N* = 2,084,721) | | Booster vaccinated group (*N* = 2,084,721) | | Risk difference per million PWs (95% CI) | rVE (95% CI) |
|---|---|---|---|---|---|---|
| | Events | Risk per million PWs | Events | Risk per million PWs | | |
| **Outcome 1: Documented SARS-CoV-2 infection** | | | | | | |
| Overall | 23,193 | 774.9 | 19,470 | 648.7 | 126.2 (125.4, 127.1) | 16.3 (14.4, 17.9) |
| Age group—no. | | | | | | |
| 60–69 years old | 16,483 | 793.7 | 14,027 | 673.6 | 120.1 (119.1, 121.1) | 15.1 (13.2, 17.0) |
| 70–79 years old | 6106 | 744.7 | 4875 | 592.6 | 152.1 (150.0, 154.1) | 20.4 (17.4, 23.4) |
| 80–89 years old | 578 | 619.6 | 544 | 582.0 | 37.6 (36, 39.2) | 6.1 (−5.5, 16.5) |
| 90 years old and above | 26 | 830.9 | 24 | 765.4 | 65.5 (52.4, 78.6) | 6.7 (−62.5, 46.4) |
| Sex—no. | | | | | | |
| Male | 11,689 | 789.9 | 9442 | 636.0 | 153.9 (152.4, 155.4) | 19.5 (17.3, 21.7) |
| Female | 11,504 | 760.2 | 10,028 | 661.0 | 99.2 (98.2, 100.1) | 13.0 (10.7, 15.3) |
| Whether having chronic disease | | | | | | |
| 0 | 14,728 | 787.3 | 12,882 | 686.9 | 100.4 (99.5, 101.3) | 12.7 (10.6, 14.8) |
| ≥1 | 8465 | 754.3 | 6588 | 585.0 | 169.3 (167.3, 171.2) | 22.5 (20.0, 25.0) |
| Time interval between 1st dose and 2nd dose | | | | | | |
| 21–56 days | 23,048 | 774.8 | 19,333 | 648.0 | 126.7 (125.9, 127.6) | 16.4 (14.8, 18.0) |
| 57–180 days | 145 | 800.3 | 137 | 754.3 | 46.0 (42.0, 49.9) | 5.9 (−18.8, 25.5) |
| 181 days and above | 0 | 0 | 0 | 0 | – | – |
| Time interval since completion of 2nd dose (days) | | | | | | |
| 181–270 | 21,549 | 795.1 | 18,172 | 668.6 | 126.6 (125.7, 127.5) | 15.9 (14.2, 17.6) |
| 271–360 | 1630 | 718.2 | 1298 | 570.2 | 147.9 (144.1, 151.8) | 20.5 (14.5, 26.1) |
| 361 and above | 14 | 25.0 | 0 | 0 | 25.0 (13.7, 35.4) | 100 (NA) |
| **Outcome 2: Severe/critical Covid-19** | | | | | | |
| Overall | 66 | 2.2 | 26 | 0.9 | 1.3 (1.1, 1.5) | 60.5 (37.8, 74.9) |
| Age group—no. | | | | | | |
| 60–69 years old | 16 | 0.8 | 9 | 0.4 | 0.3 (0.2, 0.4) | 44.4 (−25.8, 75.4) |
| 70–79 years old | 26 | 3.2 | 8 | 1.0 | 2.2 (1.7, 2.7) | 69.1 (31.9, 86.0) |
| 80–89 years old | 24 | 25.7 | 6 | 6.4 | 19.3 (14.1, 24.3) | 74.9 (38.7, 89.8) |
| 90 years old and above | 0 | 0 | 3 | 95.7 | None[a] | None[a] |
| Sex—no. | | | | | | |
| Male | 15 | 1 | 5 | 0.3 | 0.7 (0.5, 0.9) | 66.6 (8.1, 87.9) |
| Female | 51 | 3.4 | 21 | 1.4 | 2.0 (1.7, 2.3) | 58.7 (31.3, 75.1) |
| Whether having chronic disease | | | | | | |
| 0 | 29 | 1.6 | 12 | 0.6 | 0.9 (0.7, 1.1) | 58.7 (19.1, 78.9) |
| ≥1 | 37 | 3.3 | 14 | 1.2 | 2.1 (1.6, 2.5) | 61.9 (29.6, 79.4) |
| Time interval between 1st dose and 2nd dose | | | | | | |
| 21–56 days | 64 | 2.2 | 26 | 0.9 | 1.3 (1.1, 1.5) | 59.3 (35.7, 74.2) |
| 57–180 days | 2 | 11 | 0 | 0 | 11.0 (1.3, 19.6) | 100 (NA) |
| 181 days and above | 0 | 0 | 0 | 0 | – | – |
| Time interval since completion of 2nd dose (days) | | | | | | |
| 181–270 | 63 | 2.3 | 26 | 1 | 1.4 (1.2, 1.6) | 58.7 (34.7, 73.8) |
| 271–360 | 3 | 1.3 | 0 | 0 | 1.3 (0.3, 2.2) | 100 (NA) |
| 361 and above | 0 | 0 | 0 | 0 | – | – |
| **Outcome 3: Covid-19-related death** | | | | | | |
| Overall | 11 | 0.4 | 2 | 0.1 | 0.3 (0.2, 0.4) | 81.7 (17.5, 95.9) |
| Age group—no. | | | | | | |
| 60–69 years old | 3 | 0.1 | 0 | 0 | 0.1 (0, 0.2) | 100 (NA) |
| 70–79 years old | 2 | 0.2 | 1 | 0.1 | 0.1 (0, 0.2) | 50.7 (−443.5, 95.5) |
| 80–89 years old | 6 | 6.4 | 0 | 0 | 6.4 (2.4, 10.1) | 100 (NA) |
| 90 years old and above | 0 | 0 | 1 | 31.9 | None[a] | None[a] |

**Table 4 (continued) | Adjusted relative vaccine effectiveness of the booster vaccination comparing to full vaccination of inactivated Covid-19 vaccine**

| | Fully vaccinated group (N = 2,084,721) | | Booster vaccinated group (N = 2,084,721) | | Risk difference per million PWs (95% CI) | rVE (95% CI) |
|---|---|---|---|---|---|---|
| | Events | Risk per million PWs | Events | Risk per million PWs | | |
| Sex—no. | | | | | | |
| Male | 5 | 0.3 | 1 | 0.1 | 0.3 (0.1, 0.4) | 80.2 (−69.3, 97.7) |
| Female | 6 | 0.4 | 1 | 0.1 | 0.3 (0.1, 0.5) | 82.9 (−42.4, 97.9) |
| Whether having chronic disease | | | | | | |
| 0 | 7 | 0.4 | 1 | 0.1 | 0.3 (0.1, 0.5) | 85.7 (−15.9, 98.2) |
| ≥1 | 4 | 0.4 | 1 | 0.1 | 0.3 (0.1, 0.4) | 74.6 (−127.5, 97.2) |
| Time interval between 1st dose and 2nd dose | | | | | | |
| 21–56 days | 11 | 0.4 | 2 | 0.1 | 0.3 (0.2, 0.4) | 81.7 (17.5, 95.9) |
| 57–180 days | 0 | 0 | 0 | 0 | – | – |
| 181 days and above | 0 | 0 | 0 | 0 | – | – |
| Time interval since completion of 2nd dose (days) | | | | | | |
| 181–270 | 11 | 0.4 | 2 | 0.1 | 0.3 (0.2, 0.5) | 81.8 (17.7, 96.0) |
| 271–360 | 0 | 0 | 0 | 0 | – | – |
| 361 and above | 0 | 0 | 0 | 0 | – | – |

*95% CI* 95% confidence interval, *PW* person-weeks, *NA* not available.
[a]No evidence of protection based on a negative or very small positive point estimate and wide CIs.

## Design and study population

We estimated vaccine effectiveness using a matched retrospective cohort study design in which we assembled two cohorts in the target population (Fig. 2). Cohort 1 was used to compare two treatment strategies and consisted of a vaccinated group (administration of at least one dose of inactivated vaccine at the time of recruitment into the cohort) and an unvaccinated group (no administration of any Covid-19 vaccine at any time during the study follow-up period). Cohort 2 was used to compare two different treatment strategies and consisted of a booster-vaccinated group (administration of a third dose of inactivated vaccine by the time of recruitment into the study) and a fully vaccinated group (administration of two doses of inactivated vaccine by the time of study recruitment and no administration of a third dose of any COVID-19 vaccine at any time during study follow-up). Cohort 1 was used to estimate absolute VEs against Covid-19 outcomes, and cohort 2 estimated relative VEs of booster vaccination by comparing risks of Covid-19 outcomes between the booster vaccination group and the fully (2-dose) vaccinated group.

On each day of the study period, eligible individuals who received a first dose (cohort 1: vaccinated group) or received a third dose (cohort 2: booster vaccinated group) on that day were matched to eligible controls who were and remained unvaccinated (cohort 1: unvaccinated group) or had previously received two doses of inactivated vaccine but had not received a third dose (cohort 2: fully vaccinated group). The matching process was performed daily during study period. Controls in the unvaccinated group (in cohort 1) or fully vaccinated group (in cohort 2) who received a first dose of inactivated vaccine or a booster dose at a future time during the study period, respectively, were eligible to be enrolled into the vaccinated group or the booster vaccinated group, respectively. See below for handling the matched study population.

Persons meeting the following criteria were eligible for cohort 1: living in Shanghai during study period, 60 years old or older, and SARS-CoV-2-infection-free before enrollment. Individuals in cohort 1 were exact-matched in a 1:1 ratio on years-of-age, sex, presence of a chronic disease (cancer, hypertension, and type II diabetes mellitus, categorized as 0, and ≥1 comorbidities). Cohort 2 inclusion criteria were: living in Shanghai during study period, 60 years old or older, having completed two doses of inactivated vaccine at least 181 days before

enrollment between November 1, 2021 (the general recommendation date of the booster vaccination campaign in Shanghai) and July 16, 2022 (the end date of the study period), and SARS-CoV-2-infection-free before enrollment. Individuals in cohort 2 were also matched 1:1 on time interval between first dose and second dose (categorized as 21–56 days, 57–180 days, or 181 days or more), and time since receiving the second dose (categorized as 181–270 days, 271–360 days, or 361 days or more) to control for waning immunity.

Study population with a previous documented SARS-CoV-2 infection before March 25, 2021, or who had vaccine contraindications listed on the *Technical Guidelines for COVID-19 Vaccination (First Edition)* (i.e., a history of anaphylaxis to any component of the vaccine, or to the same type of vaccine; a history of severe allergic reaction to vaccines [e.g., acute anaphylactic reaction, angioedema, dyspnea]; people with uncontrolled epilepsy and other severe neurological diseases [e.g., transverse myelitis, Guillain-Barre syndrome, demyelinating diseases]; those suffering from fever, acute illness, acute onset of chronic disease, or uncontrolled severe chronic disease; pregnant women), or who had an interval between first dose and second dose within 21 days, or received third dose administration, if applicable, within 6 months after the second dose, or who received non-inactivated vaccines/heterologous inactivated vaccines during study period were excluded from the study.

Data on both individuals of a matched pair were censored once a control entered the vaccinated group (cohort 1) or the booster vaccinated group (cohort 2). Follow-up ended at the first of these events: occurrence of an outcome event, vaccination (for an unvaccinated group study population), vaccination of the matched control (for a vaccinated group study population), booster vaccination (for a fully vaccinated group study population), booster vaccination of the matched control (for a booster vaccinated group study population), or the end of study period (July 16, 2022).

## Outcomes

Three primary outcomes were documented SARS-CoV-2 infection, severe/critical Covid-19, and Covid-19-related death. Details of Covid-19 infection severity levels and assessments are published elsewhere[12]. Severe Covid-19 must meet any of the following criteria: (a) respiratory distress (respiration rate [RR] ≥ 30 breaths per

min), (b) oxygen saturation ≤93% at rest, or (c) arterial partial pressure of oxygen/fraction of inspired oxygen ≤300 mmHg. Additionally, cases with chest imaging that shows obvious lesion progression within 24–48 h >50% were managed and considered as severe Covid-19. Critical Covid-19 must meet any of the following criteria: (a) respiratory failure requiring mechanical ventilation; (b) shock, or (c) with other organ failure that requires ICU care. Covid-19-related death was assessed by medical institutions. For each matched pair, follow-up started on the 8th day (7-day cutoff) after receiving a first dose (cohort 1: vaccinated group) or a booster dose (cohort 2: booster vaccinated group). Follow-up ended after the occurrence of any of the three study outcomes or a censoring event, whichever happened first. The 7-day cutoff was to ensure sufficient time to induce an immune response.

### Statistical analysis

Non-normally distributed continuous variables were expressed as medians [interquartile ranges, IQR], and categorical variables were expressed as counts and proportions. Groups were compared with standardized mean differences (SMD), with a value of less than 0.1 indicating adequate matching. We constructed cumulative incidence curves of primary outcomes in each group using the Kaplan–Meier estimator. Risks were compared via ratios and differences. Hazard ratios (HRs) for between-group comparisons of incidence and their corresponding 95% confidence intervals were estimated with Cox regression, adjusted for whether having hypertension (binary variable, yes or no), whether having diabetes (binary variable, yes or no), and whether having cancer (binary variable, yes or no). Vaccine effectiveness was 1−HR, and relative VE was calculated as 1-HR (3-dose group vs 2 dose group in cohort 2). We estimated risk ratios for each outcome using only matched pairs in which both individuals were still at risk 7 days after receipt of the first vaccine dose (the matching date) for those vaccinated. We analyzed outcomes in the full population and in subgroup strata of age, sex, presence or number of chronic diseases, time between first dose and second dose (cohort 2), and time between full vaccination and a booster dose (cohort 2). Schoenfeld residuals and time-dependent coefficient plots were used to test the proportional-hazards assumption. We extracted two subsets from cohort 1 to estimate VEs of at least two-dose vaccination, and three-dose vaccination against Covid-19 outcomes.

We also did sensitivity analyses to cohort 1 and cohort 2 dataset with Negative binomial regression to estimate incidence rate ratios (IRRs). In another sensitivity analyses, we made analyses without restriction to follow-up days (from time 0 days), and we selected controls with sampling without replacement on same matching criteria. Analyses were performed with R version 4.1.3.

### Reporting summary

Further information on research design is available in the Nature Portfolio Reporting Summary linked to this article.

## Data availability

The dataset from this study is held in coded form at Shanghai Center for Disease Control and Prevention. While legal data sharing agreements between Shanghai CDC and its superior departments in charge (e.g., Health Commission or local government) prohibit Shanghai CDC from making the dataset publicly available. Access may be granted to those who through a request with specific data needs, analysis plans, and dissemination plans to Zhuoying Huang (e-mail: huangzhuoying@scdc.sh.cn), Dr. Xiaodong Sun (e-mail: sunxiaodong@scdc.sh.cn), and Dr. Weibing Wang (e-mail: wwb@fudan.edu.cn). The authors will give feedback within 30 days. However, individual identification information may not be available for public use.

## Code availability

The data were analyzed in R version 4.1.3 using the following packages: survival (3.3-1), and survminer package. The computer code used to generate the results reported in this study are available from the corresponding author upon request.

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

## Acknowledgements

This work was supported by the Science and Technology Commission of Shanghai Municipality (Grant No. 22YJ1400200, Grant No. 20JC1410200), Shanghai New Three-year Action Plan for Public Health (2023-2025), Shanghai "Rising Stars of Medical Talents" Youth Development Program, Youth Medical Talents-Public Health Leadership Program (Grant No. SHWSRS (2020)87), and Shanghai Municipal Science and Technology Major Project (Grant No. ZD2021CY001). We sincerely appreciate the valuable comments and suggestions from Professor Lance Rodewald from Institute of Immunization, Chinese Center for Disease Control and Prevention, who allowed us to greatly improve the quality of the manuscript. Science and Technology Commission of Shanghai Municipality.

## Author contributions

Z.H. conceptualized the study, collected data, and drafted the manuscript. S.X. performed analysis and drafted the manuscript. JC.L., J.Q., N.W., F.T., and D.L. performed data clean and controlled quality. L.W. assisted with study design. J.R., Z.L., Y.W., and J.L. collected data. X.G. and J.C. reviewed manuscript. X.S. conceptualized the study, assisted with the analysis, and reviewed and edited the manuscript. W.W. assisted with the analysis, and reviewed and edited the manuscript. All authors read and approved the final manuscript.

## Competing interests

The authors declare no competing interests.

## Ethics approval

The study was approved by the Ethical Review Committee in the Shanghai Center for Disease Control and Prevention (approval number: 2022–20).
