## [Peer Review File · Nature Communications]

Effectiveness of inactivated COVID-19 vaccines among older adults in Shanghai: retrospective cohort studyREVIEWER COMMENTS

Reviewer #1 (Remarks to the Author):

The authors performed two matched cohorts to evaluate the vaccine effectiveness of inactivated vaccines in Shanghai during 2021 and 2022, among the general population aged ≥ 60 y. One of the strengths is the application of a target emulated trial. Some limitations are important to highlight, such as a short follow-up time to evaluate 1 dose VE (~ 3 months on average), which is about the peak of protection from inactivated vaccines, somehow overoptimistic and imprecise VE compared to the literature, lack of adjustment by VoC and other confounding factors, and lack of power for some outcomes.

Please, find below some concerns. Many thanks for the opportunity to read your work.

Major

1 - Please, better explain the exclusion criteria. Specifically, "Potential subjects with a previous documented SARS-CoV-2 infection, or who did not follow a recommendation vaccination schedule".

- a. What the authors meant by "potential subjects"? And what is defined as
- b. "did not follow a recommendation vaccination schedule? How this bias the results?"

2 - To run a target emulated trial, the authors must follow a RCT inclusion/exclusion criteria would be. Some characteristics of it seem to not be followed or it was not clearly written.

a. Take the example of eligibility to be randomized. It is not clear, since in line 85, the unvaccinated were those never vaccinated, but it is not the same what is written in paragraph line 92. When the authors conditioned those who had never been vaccinated in the unvaccinated group, they conditioned in the future, on being never vaccinated. Therefore, this fails the eligibility of being randomized. The authors must run a rolling entry cohort, and so, allow to be analysed in the control group. Following the methods, it seems a proper rolling cohort was conducted. Please clarify.

b. Criteria for cohort 1, lines 101-104, is not clear too. How can someone be eligible and receipt of an inactivated vaccine. The authors are merging inclusion criteria in the cohort with the exposure.

c. How the authors managed those that died by other causes during the follow-up? Being a cohort of older, this might not be a dismissable competing event.

3 - Please, provide a brief definition of severe/critical COVID-19 and death in this manuscript. This is key to the reader.

4 - The 7 days period (or 14 days) is a good opportunity to evaluate bias in the estimate (10.1097/EDE.0000000000001484). Instead of taking that out the authors could provide the reader what VE was estimated in this period, so giving a clue on bias.

5 - It is not clear which is the source list to obtain all elderly living in Shanghai in the period. Does the vaccination program has the list of everyone living in Shanghai?

6 - Please, describe the sensitivity analyses in the methods.

The analysis described in Table S3 and S4 are expected to be biased, since they are not compliant with the target trial. This reviewer suggests to exclude them. Please, also better describe about the subsets analyses.

7 - Was the matching done with replacement? How from ~ 5 million individuals we have 1:1 matching? Was this because of censoring and moving controls to the vaccinated pool? Which impact has the replacement on the standard errors? How many vaccinated were not matched?

8 - According the COVID-19 features, and WHO guidance on VE, control for regional characteristics

is necessary. Any data to control for neighbourhood or other spatial feature?

9 - Following item 9, time is also crucial. The cohort is build on calendar time, but there is VoC time. How this was handled?

10 - Why did the authors choose to evaluate VE only after 1 dose? This is not usual and shows partially the whole history. This reviewer strongly suggest to evaluate 2nd dose, as well as, to open the time since the vaccination.

a - The estimated VE are quite optimistic based on the published data. Even if we consider Omicron was on the analysed data. Look at COVID-19 death among the most elderly, or with comorbidities. This is not in accordance to the published literature if we analyse even two doses, if we look at 1 dose, the current estimated are fairly far from the literature. It is close to the Jara paper, but with a limited follow-up time and not covering any major outbreak in Chile, different from this analysis. Additionally, the fact of VE be similar among those with and without comorbidities, makes a red flag on the internal validity of the analysis.

10.1016/S2666-7568(22)00035-6

10.1016/j.lana.2022.100296

10.1136/bmj-2022-070102

10.1136/bmj.n2015

11 - Please, state in the abstract and conclusions these VE are valid in a scenario of zero-COVID and other NPI policies, extensive testing, which impact on VE estimates.

a - The force of infection affects the VE, not only the by mechanism stated in the limitations(10.1038/s41541-021-00316-5)

12 - If a non proportional cox was observed, and so a time-varying Cox was fit, how the authors get a single point-estimate for VE? Please, clarify.

13 - Could the authors show and potentially analyse VE by vaccine brand?

ABSTRACT: Please, revise. Currently, it is hard to read and follow.

Minor

1 - Please, update the numbers in the introduction and replace "to date" to the actual month, since it is a dynamic value (lines 28-30).

2 - Please, temper the statement that inactivated vaccines provide strong protection against severe or fatal illness. It depends heavily on age, time from vaccine shot and VoC (lines 39-42)

10.1016/S2666-7568(22)00035-6

10.1016/j.lana.2022.100296

10.1136/bmj-2022-070102

10.1136/bmj.n2015

3 - Line 150: analyses were conducted in R, since RStudio is just an IDE.

Reviewer #2 (Remarks to the Author):

This paper describes an important study of the effectiveness of different number of doses of inactivated COVID-19 vaccines in Shanghai, China between March and July, 2022. The study is original and important because VE data from China have been so limited due to a lack of SARS-CoV-2 circulation. That situation is changing rapidly and so reports on the performance of vaccines are policy-relevant and timely. The study design seems appropriate (if imperfect) to respond to

the question at hand and I commend the authors for a diligent and comprehensive work.

Unfortunately, the manuscript requires major revisions before publication. My main concern is about clarity and methodological accuracy and the current conclusions do not adequately reflect uncertainty and may therefore be misleading. The writing is sometimes confusing and/or repetitive which hinder understanding. Major comments:

Introduction

- There is inadequate context on COVID-19 and related control measures in Shanghai. During the study period I guess SARS-CoV-2 circulation was minimal; I guess lockdowns were standard; and I guess cases were mostly ascertained from mandatory PCR screening. We need also to know what kind of NPIs were in place over the study period and how they may have impacted transmission and the likelihood to vaccinate.
- We also need to know how the vaccination campaigns were conducted; and any information possible about the reasons for the low VCR in the elderly. Were there any vaccine mandates or inducements? Was there widespread fear of vaccines?
- It would be helpful to see an epi curve; and a graph of the evolving vaccination coverage in Shanghai; over time, indicating when the study took place.

Methods

- The term "to emulate clinical trials" is used several times and should be removed. This is a classical observational study design, all of which would hope to emulate clinical trials, but this study has no more of a claim to draw causal conclusions than any other (in fact due to the minimal control for confounders, it is liable to residual confounding).
- The design is a little confusing and I worry it's not necessary. There are two distinct cohorts with overlapping populations and many within cohort comparisons. But the source population for all is the same. Is there a reason authors cannot draw one cohort with different exposure status (zero; one; two; three doses)?
- The booster cohort matches with individuals vaccinated with a second dose long ago and whose immune status may have waned to baseline levels. It's not really a 3 vs 2 dose comparison; it's a "recent 3-dose" vs "waned 2-dose" comparison. A more valid (and interesting) comparison would be with those receiving the second dose on the same day. Maybe that wouldn't be possible but this should be carefully described so there is no ambiguity.
- Matching scheme: due to the focal nature of COVID, geography is important. Is there no way to match (or adjust) on geography?
- "Adjusted" results are presented but there's no mention of this in the methods. Controlling for matched variables can be problematic and I think should be the subject of a sensitivity analysis.
- How was rVE calculated? Not mentioned in the methods
- It would be good to know what proportion of cases reported symptoms – is this available?

Results

- Table 1 shows well-balanced cohorts. But most of these variables are matched so it's not really necessary to say "variables were well balanced between the study groups". It's a design feature.
- There are so many comparisons which are confusing. There is VE of a mixed group (1, 2 or 3 doses) vs no vaccine group; and a comparison of 3 vs 2 doses (long ago); and a separate re-analysis of cohort 1 to split out the 3-dose group to compare to the unvaccinated. Why not make 1 big cohort and compare within the exposure categories, and time?
- Do you have information on how many cases experienced symptoms; and how many were picked up during mandatory screening? This would be important for interpretation.
- Tables: you call VE "1-adjusted HR" but VE was already defined in the methods. Easier to just call it "VE"

Discussion

- It's really essential for interpretation to provide some understanding of the intensity and characteristics of COVID-19 in Shanghai over the study period. Recommend moving the information which is provided (lines 313 – 316) to the introduction; with some more detail.
- You talk about the low CFR. Probably, this is because cases were ascertained through mass screening and therefore had a milder average clinical severity than other countries.
- Lines 293 – 296. Talks about "a guarantee of similar exposure risk" but this is untrue. It's true

that mandatory screening may improve case ascertainment, leading to identification of a high proportion of infections – this is a real strength of the study. But there is no guarantee that vaccinated/unvaccinated people behave in a similar way or arise from the same source populations. For geographical, demographic or other reasons those populations maybe very different. You cannot account for this through adjustment in the study because the data are not available. This is an important limitation of the study which is mentioned, but only briefly. The negative VEs observed in table 2 may be caused by this (if vaccinated people experience more exposure than their unvaccinated peers).

In summary, I think this study has a lot of potential and I commend the author for their work. A simpler and clearer analysis is needed; with more explanation of limitations. I would also like to see recommendations about the frequency of boosting – you have data to describe VE as a function of time since last dose. This can be very important for control of the current Chinese epidemic.

Response to Reviewers

We thank the Editor and the two anonymous reviewers for their constructive inputs to our paper. We have addressed all the comments. Please find our point-to-point responses below.

Reviewer #1:

The authors performed two matched cohorts to evaluate the vaccine effectiveness of inactivated vaccines in Shanghai during 2021 and 2022, among the general population aged ≥ 60 y. One of the strengths is the application of a target emulated trial. Some limitations are important to highlight, such as a short follow-up time to evaluate 1 dose VE (~ 3 months on average), which is about the peak of protection from inactivated vaccines, somehow overoptimistic and imprecise VE compared to the literature, lack of adjustment by VoC and other confounding factors, and lack of power for some outcomes. Please, find below some concerns. Many thanks for the opportunity to read your work.

Response: Thanks for this general comments.

1. Please, better explain the exclusion criteria. Specifically, "Potential subjects with a previous documented SARS-CoV-2 infection, or who did not follow a recommendation vaccination schedule".

1a. What the authors meant by "potential subjects"? And what is defined as

Response: Thanks for this suggestion. To make the definition of "potential subjects" clearly, we have revised this sentence as following (page 6, line 262-272):

Subjects with a previous documented SARS-CoV-2 infection before March 25, 2021, or involving vaccination contraindications listed on the Technical Guidelines for COVID-19 Vaccination (First Edition) (specifically, a history of anaphylaxis to any component of the vaccine, or to the same type of vaccine; a history of severe allergic reactions to vaccines (like acute anaphylactic reactions, angioedema, dyspnea, etc.); people with uncontrolled epilepsy and other severe neurological diseases (e.g. Transverse myelitis, Guillain-Barre syndrome, demyelinating diseases, etc.); those suffering from fever, acute illness, acute onset of chronic disease, or uncontrolled severe chronic disease; pregnant women), or who did not follow a recommended vaccination schedule (the time interval between the first dose and second dose is 21 to 56 days, and a third dose

is administered 6 months after the second dose), or receiving non-inactivated vaccines/heterologous inactivated vaccines during study period were excluded from the study.

1b. "did not follow a recommendation vaccination schedule? How this bias the results?"

Response: Recommended vaccination schedule has been put in the method section (page 6, line 269-272). Following a recommended vaccination schedule is important to ensure the efficacy of vaccines from reproducible immune system responses. Allowing use of off-schedule vaccinations in the study would have potential of introducing bias into the VE results compared with using the vaccines as recommended. Referencing questions and answers from WHO (<https://www.who.int/news-room/questions-and-answers/item/vaccines-and-immunization-what-is-vaccination>) and introduction from CDC's website (*Reasons to Follow CDC's Recommended Immunization Schedule*, <https://www.cdc.gov/vaccines/parents/schedules/reasons-follow-schedule.html>). In addition to safety, vaccines given on their recommended schedule provide people promising protection. When the doses are timely, the vaccine itself can have the best effect. People who are not vaccinated on schedule are more vulnerable to infection than those following schedule. If our study included such participants, the results might bias the estimation of VE towards null hypothesis.

2. To run a target emulated trial, the authors must follow a RCT inclusion/exclusion criteria would be. Some characteristics of it seem to not be followed or it was not clearly written.

2a. Take the example of eligibility to be randomized. It is not clear, since in line 85, the unvaccinated were those never vaccinated, but it is not the same what is written in paragraph line 92. When the authors conditioned those who had never been vaccinated in the unvaccinated group, they conditioned in the future, on being never vaccinated. Therefore, this fails the eligibility of being randomized. The authors must run a rolling entry cohort, and so, allow to be analysed in the control group. Following the methods, it seems a proper rolling cohort was conducted. Please clarify.

Response: Thanks for raising this important point. As you mentioned, our study is not a strict-defined RCT, and randomization is not applicable. We assembled two retrospective cohorts to emulate clinical trials in the target population to estimate VEs (References: 10.1056/NEJMoa2101765, 10.1056/NEJMoa2200797, and

10.1016/S1473-3099(22)00292-4). Another reviewer for this manuscript thought the study design seemed appropriate (if imperfect) to respond to the question at hand and commend us for a diligent and comprehensive work.

For the two cohorts in our study, controls in cohort 1 were those never vaccinated and in cohort 2 were those who were administered two doses of inactivated vaccine. When controls in the unvaccinated group (in cohort 1) received the first dose of inactivated vaccine at a future time during the study period, they were eligible to be enrolled into the vaccinated group; when controls in the 2-dose group (in cohort 2) received a booster dose at a future time during the study period, they were eligible to be enrolled into the booster vaccinated group. However, such controls and their matched individuals were censored in the analysis and they didn't contribute to the study outcomes. In reality, this was a rolling entry cohort, and the matching process in our study was performed daily.

2b. Criteria for cohort 1, lines 101-104, is not clear too. How can someone be eligible and receipt of an inactivated vaccine. The authors are merging inclusion criteria in the cohort with the exposure.

Response: Thanks for your comment. The following sentences explained how someone can be eligible and receive an inactivated vaccine (page 6, line 262-268):

...or who had vaccine contraindications listed on the Technical Guidelines for COVID-19 Vaccination (First Edition) (i.e, a history of anaphylaxis to any component of the vaccine, or to the same type of vaccine; a history of severe allergic reaction to vaccines [e.g., acute anaphylactic reaction, angioedema, dyspnea]; people with uncontrolled epilepsy and other severe neurological diseases [e.g. transverse myelitis, Guillain-Barre syndrome, demyelinating diseases]; those suffering from fever, acute illness, acute onset of chronic disease, or uncontrolled severe chronic disease; pregnant women)... during study period were excluded from the study.

Thank you for your reminder. We have made clearer descriptions of inclusion criteria (page 6, line 251-258), exclusion criteria (page 6, line 262-272), and exposure measurement (page 6, line 242-245) in the manuscript.

2c. How the authors managed those that died by other causes during the follow-up? Being a cohort of older, this might not be a dismissable competing event.

Response: A total of 21636 deaths (including Covid-19 related death) occurred in the cohort between March 2021 and June 2022. Those that died by other causes during the follow-up were treated as censored. In consideration of the low cumulative incidence of SARS-CoV-2 infection (2.6%) and the low CFR (0.2%) of Covid-19 related death during the outbreak between March 2022 and Jul 2022 in Shanghai, even if we assumed that who died were still alive, their probability of having study outcomes was quite low. Against SARS-CoV-2 infection, VE of receiving one or more doses inactivated vaccines was 21.4% (95%CI: 20.0–22.7) in competing risk analysis, which is similar to VE in our primary analysis (Table 2, 21.6% (95%CI: 20.2–23.0)).

3. Please, provide a brief definition of severe/critical COVID-19 and death in this manuscript. This is key to the reader.

Response: We appreciate this request. Severity assessment criteria in our study was in accordance with the *Diagnosis and Treatment Protocol for COVID-19 (Trial Version 9)*. Severe Covid-19 must meet any of the following criteria: a) respiratory distress (Respiration Rate [RR] \geq 30 breaths per min), b) oxygen saturation \leq 93% at rest, c) arterial partial pressure of oxygen/fraction of inspired oxygen \leq 300mmHg. Additionally, cases with chest imaging that shows obvious lesion progression within 24-48 hours $>$ 50% shall be managed as severe Covid-19. Critical Covid-19 must meet any of the following criteria: a) respiratory failure and requiring mechanical ventilation; b) shock, c) with other organ failure that requires ICU care. COVID-19 related death is assessed by medical institutions.

We have added this definition into the methods section (page 6-7, line 281-289).

4. The 7 days period (or 14 days) is a good opportunity to evaluate bias in the estimate (10.1097/EDE.0000000000001484). Instead of taking that out the authors could provide the reader what VE was estimated in this period, so giving a clue on bias.

Response: In cohort 1, 262808 vaccinated individuals who followed up within 7 days were excluded from the primary analysis. For these individuals, VEs against SARS-CoV-2 infection, severe/critical Covid-19, and Covid-19 related death were 100%.

To ensure validity, WHO and many references exclude persons vaccinated within approximately 7 days or 14 days after the first dose from the primary analysis outcomes, as the individual's immunization status when they were infected may be uncertain.

5. It is not clear which is the source list to obtain all elderly living in Shanghai in the period. Does the vaccination program have the list of everyone living in Shanghai?

Response: During the study period, 25.18 million people living in Shanghai received 4 rounds of city-wide nucleic acid amplification testing (NAAT), and RT-PCR testing was universal regardless of COVID-19 associated symptoms. We obtained this list of everyone living in Shanghai, then retrieved their vaccination history from the Shanghai Group Immunization System and National Immunization Program Information System.

6. Please, describe the sensitivity analyses in the methods.

The analysis described in Table S3 and S4 are expected to be biased, since they are not compliant with the target trial. This reviewer suggests to exclude them. Please, also better describe about the subsets analyses.

Response: Initially we emulated an RCT with the analysis described in Table 2-4 as an intention-to-treat analysis, and the analysis in Table S3 and S4 as a per-protocol analysis. In consideration of the reviewer's suggestion, we have removed these two tables.

7. Was the matching done with replacement? How from ~5 million individuals we have 1:1 matching? Was this because of censoring and moving controls to the vaccinated pool? Which impact has the replacement on the standard errors? How many vaccinated were not matched?

Response: Yes, the controls were randomly selected with replacement (see Fig. 1). This matching process required 10 days on an SQL server. In cohort 1, all vaccinated individuals were matched, and 783330 individuals were repeatedly selected as controls in the sampling with replacement. In cohort 2, 238 booster vaccinated individuals were not matched (and excluded from analysis), and 518714 were repeatedly selected as controls in the sampling with replacement.

The standard error of sampling without replacement is always smaller than that of sampling with replacement. If the population is very large, sampling with replacement isn't much different from sampling without replacement. Referring to 10.1016/S1473-3099(22)00292-4, VE and 95%CI under sampling with replacement (52.5% (51.3–53.7)) were similar to that under sampling without replacement (52.0% (50.8–53.2)). The population size of control source in our study is quite large (5.4 million), thus we

believe there is little impact of the replacement on the standard errors.

8. According the COVID-19 features, and WHO guidance on VE, control for regional characteristics is necessary. Any data to control for neighborhood or other spatial feature?

Response: We appreciate your comment. Unfortunately, we were unable to control for neighborhood or other spatial features, and have added this point to limitation part as follows (page 5, line 202-204):

Some key variables, such as occupational exposure, neighborhood or other spatial feature, living environment (community or nursing home), were not available and may not have been balanced, leading to an unknown bias in an unknown direction.

9. Following item 9, time is also crucial. The cohort is built on calendar time, but there is VoC time. How this was handled?

Response: In our study, a total of 50139 cases emerged during study period. Due to the fact that sequencing results of virus isolates from 129 COVID-19 patients between late February 2022 and May 2022 showed that Omicron BA. 2 was the dominate sub-lineage in Shanghai (Reference: 10.1016/S0140-6736(22)00838-8), we assumed they were related to Omicron variant.

Temporal distribution of Cases during study period	
Time period	No. of cases (%)
Feb, 2022	6 (0.01)
Mar, 2022	569 (1.13)
Apr, 2022	44261 (88.28)
May, 2022	5245 (10.46)
Jun, 2022	43 (0.09)
Jul, 2022	15 (0.03)

10. Why did the authors choose to evaluate VE only after 1 dose? This is not usual and shows partially the whole history. This reviewer strongly suggests to evaluate 2nd dose, as well as, to open the time since the vaccination.

Response: Actually, VEs only after 1 dose were not evaluated separately. We analyzed VEs of a mixed group (1-dose, 2-dose, and 3-dose, described in Table 2), and time-varying VEs of the 2nd dose and 3rd dose, respectively (Table 3).

In cohort 1, 262808 vaccinated individuals were excluded from the primary analysis for violating the recommended vaccination schedule. For these population, VEs of 2-

dose against SARS-CoV-2 infection, severe/critical Covid-19, and Covid-19 related death were 99.8% (95% CI: 98.7-100.0), 100% (NA), and NA, respectively. VEs of 3-dose against SARS-CoV-2 infection, severe/critical Covid-19, and Covid-19 related death were 100% (95% CI: 99.9-100.0), 100% (NA), and 100% (NA), respectively. To ensure validity, we didn't put these analyses in manuscript.

11. The estimated VE are quite optimistic based on the published data. Even if we consider Omicron was on the analysed data. Look at COVID-19 death among the most elderly, or with comorbidities. This is not in accordance to the published literature if we analyse even two doses, if we look at 1 dose, the current estimated are fairly far from the literature. It is close to the Jara paper, but with a limited follow-up time and not covering any major outbreak in Chile, different from this analysis. Additionally, the fact of VE be similar among those with and without comorbidities, makes a red flag on the internal validity of the analysis.

10.1016/S2666-7568(22)00035-6

10.1016/j.lana.2022.100296

10.1136/bmj-2022-070102

10.1136/bmj.n2015

Response: Thanks. On one hand, optimistic VEs might attribute to the massive NAAT, which helped early discovery, diagnosis and treatment for infection. On the other hand, the inactivated COVID-19 vaccine showed good immunogenicity and safety in patients aged ≥ 60 years suffering from hypertension or(/and) diabetes mellitus (reference: 10.3390/vaccines10071020). In addition, our study period covered the last outbreaks (Mar 2022 to May 2022) in Shanghai, and the follow-up time was longer than or similar to some published papers (references: 10.1016/S1473-3099(22)00292-4, 10.1056/NEJMoa2200797, 10.1016/S0140-6736(21)02183-8, 10.1016/j.medj.2021.06.007).

12. Please, state in the abstract and conclusions these VE are valid in a scenario of zero-COVID and other NPI policies, extensive testing, which impact on VE estimates. The force of infection affects the VE, not only the by mechanism stated in the limitations (10.1038/s41541-021-00316-5)

Response: Thanks a lot for you comment, and we have revised abstract and conclusions as following:

Abstract:

... Although our study was conducted under a scenario of persistent dynamic zero-Covid policy and non-pharmaceutical interventions, there is no doubt that promoting high uptake of the full vaccination series with booster dose administration among adults 60 years of age or older is critically important, especially for vulnerable individuals with coexisting medical conditions.

Conclusions:

Although our study was conducted under a scenario of persistent dynamic zero-Covid policy and non-pharmaceutical interventions, the findings still have clear implications for the current vaccination strategy in China in the context of the current global Covid-19 epidemic: it is critically important to promote high uptake of the full vaccination series with booster dose administration among adults 60 years of age or older, especially for vulnerable individuals with coexisting medical conditions.

13. If a non proportional cox was observed, and so a time-varying Cox was fit, how the authors get a single point-estimate for VE? Please, clarify.

Response: We appreciate this question. Non-proportional cox was observed in subset analysis (Table 3), rather than in the main analysis (Table 2 and 4), as we now describe in the methods section (page 7, line 307-311)

14. Could the authors show and potentially analyse VE by vaccine brand?

Response: Thanks a lot for your comment. VEs evaluated by vaccine brand were similar, thus we didn't put them in the manuscript.

15. ABSTRACT: Please, revise. Currently, it is hard to read and follow.

Response: Thanks, we have revised the abstract accordingly.

16. Please, update the numbers in the introduction and replace "to date" to the actual month, since it is a dynamic value (lines 28-30).

Response: Thanks. We have replaced "to date" to "Through January 10 2023", and updated the statistic.

17. Please, temper the statement that inactivated vaccines provide strong protection against severe or fatal illness. It depends heavily on age, time from vaccine shot and

VoC (lines 39-42)

10.1016/S2666-7568(22)00035-6

10.1016/j.lana.2022.100296

10.1136/bmj-2022-070102

10.1136/bmj.n2015

Response: Thanks. We have replaced “strong protection” with “promising results” (page 2, line 38).

18. Line 150: analyses were conducted in R, since RStudio is just an IDE.

Response: Thanks a lot for your kindly reminder, and we have changed “RStudio 2022.02.3+492” to “R version 4.1.3” (page 7, lines 311).

Reviewer #2:

This paper describes an important study of the effectiveness of different number of doses of inactivated COVID-19 vaccines in Shanghai, China between March and July, 2022. The study is original and important because VE data from China have been so limited due to a lack of SARS-CoV-2 circulation. That situation is changing rapidly and so reports on the performance of vaccines are policy-relevant and timely. The study design seems appropriate (if imperfect) to respond to the question at hand and I commend the authors for a diligent and comprehensive work. Unfortunately, the manuscript requires major revisions before publication. My main concern is about clarity and methodological accuracy and the current conclusions do not adequately reflect uncertainty and may therefore be misleading. The writing is sometimes confusing and/or repetitive which hinder understanding.

Response: We thank the reviewer for the constructive inputs to our paper.

1. There is inadequate context on COVID-19 and related control measures in Shanghai. During the study period I guess SARS-CoV-2 circulation was minimal; I guess lockdowns were standard; and I guess cases were mostly ascertained from mandatory PCR screening. We need also to know what kind of NPIs were in place over the study period and how they may have impacted transmission and the likelihood to vaccinate.

Response: We appreciate the suggestion. We have added following sentences to the introduction (page 2, line 55-60):

...and the citywide vaccination campaign was suspended during the outbreak. The epidemic prevention and control strategy of Covid-19 changed from a targeted approach (March 2021 to March 2022), to strict non-pharmaceutical interventions (NPIs), such as citywide home quarantine, massive nucleic acid amplification testing/rapid antigen testing (NAAT/RAT), and centralized quarantine of close contacts (March 2022 to May 2022), to regular periodic routine NAAT (May 2022 to Dec 7 2022).

2. We also need to know how the vaccination campaigns were conducted; and any information possible about the reasons for the low VCR in the elderly. Were there any vaccine mandates or inducements? Was there widespread fear of vaccines?

Response: Vaccination campaigns were voluntary and with informed consent. Our previous work (10.3390/vaccines10010091) showed the willingness to vaccinate the older in their family was around 40% in Shanghai. Recommendations from the

government, doctors, friends, and relatives increased the acceptance of vaccination. If the government and doctors promote the efficacy and safety of vaccines through social media and emphasize the importance of universal vaccination, the public acceptance of vaccination increases.

3. It would be helpful to see an epi curve; and a graph of the evolving vaccination coverage in Shanghai; over time, indicating when the study took place.

Response: Thanks for your advice. Following picture displayed an epi curve and vaccination coverage among elderly in Shanghai.

Methods

4. The term “to emulate clinical trials” is used several times and should be removed. This is a classical observational study design, all of which would hope to emulate clinical trials, but this study has no more of a claim to draw causal conclusions than any other (in fact due to the minimal control for confounders, it is liable to residual confounding).

Response: We agree with the reviewer that our analysis is liable to confounding caused by other potential factors, although we have controlled age, sex, kinds of chronic diseases, and time since completion of 2-dose vaccine schedule. As you suggested, we have removed the term “emulate clinical trials” in main text except for the Method section, and the abstract.

5. The design is a little confusing and I worry it's not necessary. There are two distinct cohorts with overlapping populations and many within cohort comparisons. But the source population for all is the same. Is there a reason authors cannot draw one cohort with different exposure status (zero; one; two; three doses)?

Response: Yes, the source population in our analysis is the same. Cohort 1 assessed VEs of mixed vaccination status, and cohort 2 assessed relative VE of 3rd dose comparing to 2nd dose. Drawing one cohort where 0-, 1-, 2-, and 3-dose groups were exactly matched in a ratio of 1:1:1:1 is a complex and challenging process.

6. The booster cohort matches with individuals vaccinated with a second dose long ago and whose immune status may have waned to baseline levels. It's not really a 3 vs 2 dose comparison; it's a "recent 3-dose" vs "waned 2-dose" comparison. A more valid (and interesting) comparison would be with those receiving the second dose on the same day. Maybe that wouldn't be possible but this should be carefully described so there is no ambiguity.

Response: Thanks. In our matching process of cohort 2, participants in 3-dose group and 2-dose group had same time interval between entering the study and completing the 2nd dose. Although not on the same day, they received the 2nd dose in a same time range, which made a comparable protection from 2-dose vaccination and could evaluate the relative VEs of 3rd dose objectively.

7. Matching scheme: due to the focal nature of COVID, geography is important. Is there no way to match (or adjust) on geography?

Response: We agree with your point. However, we did not have geographic features to be controlled or matched for. We have added this point to the limitations as follows (page 5, line 202-204):

Some key variables, such as occupational exposure, neighborhood or other spatial feature, living environment (community or nursing home), were not available and may not have been balanced, leading to an unknown bias in an unknown direction.

8. "Adjusted" results are presented but there's no mention of this in the methods. Controlling for matched variables can be problematic and I think should be the subject of a sensitivity analysis.

Response: Thanks. We have re-analyzed following your suggestion, and updated

analysis results.

9. How was rVE calculated? Not mentioned in the methods

Response: “rVE” in this manuscript referred to VEs of 3-dose vaccination comparing to 2-dose vaccination of inactivated Covid-19 vaccine. We have added following sentence to methods section (page 6, line 237-239):

Cohort 1 was used to estimate absolute VEs against Covid-19 outcomes, and cohort 2 estimated relative VEs of booster vaccination by comparing Covid-19 outcomes between the booster vaccination group and the fully (2-dose) vaccinated group.

10. It would be good to know what proportion of cases reported symptoms – is this available?

Response: This is a good suggestion. However, we did not have access to data from individual epidemiological investigations, therefore we could not report the proportion of cases reporting symptoms.

11. Table 1 shows well-balanced cohorts. But most of these variables are matched so it's not really necessary to say “variables were well balanced between the study groups”. It's a design feature.

Response: We have removed this sentence.

12. There are so many comparisons which are confusing. There is VE of a mixed group (1, 2 or 3 doses) vs no vaccine group; and a comparison of 3 vs 2 doses (long ago); and a separate re-analysis of cohort 1 to split out the 3-dose group to compare to the unvaccinated. Why not make 1 big cohort and compare within the exposure categories, and time?

Response: Thanks for your comment. We assembled two matched cohorts (in a ratio of 1:1) to estimate VEs as you listed above. Assembling one big cohort also could estimate above-mentioned VEs. However, assembling one big cohort where unvaccinated, 1-, 2-, and 3-dose groups were exactly matched and exposure status (vaccination status) is time-varying is a considerable challenge. If we did not perform matching, imbalance and incomparability across four groups would have been inevitable. The design of two cohorts in our study seems complex, but the matching and analysis process are more straightforward in this design.

13. Do you have information on how many cases experienced symptoms; and how many were picked up during mandatory screening? This would be important for interpretation.

Response: As mentioned above we did not have information on the number of cases experiencing symptoms. However, we do have information on cases picked up during massive screening – shown below.

Time period	No. of cases (%)
Feb, 2022	6 (0.01)
Mar, 2022	569 (1.13)
Apr, 2022	44261 (88.28)
May, 2022	5245 (10.46)
Jun, 2022	43 (0.09)
Jul, 2022	15 (0.03)

14. Tables: you call VE “1-adjusted HR” but VE was already defined in the methods. Easier to just call it “VE”

Response: Thanks, we have replaced “1-adjusted HR” to “VE” in the tables.

15. It’s really essential for interpretation to provide some understanding of the intensity and characteristics of COVID-19 in Shanghai over the study period. Recommend moving the information which is provided (lines 313 – 316) to the introduction; with some more detail.

Response: Thanks. We have moved these sentences to the introduction with more details.

16. You talk about the low CFR. Probably, this is because cases were ascertained through mass screening and therefore had a milder average clinical severity than other countries.

Response: We agree with you that the low CFR were partly due to the mass screening, and have added a sentence into discussion section as following (page 4, line 138-140):
...In addition, massive screening in strict lockdown period (Mar 30, 2022 to May 31, 2022) achieved early discovery, diagnosis and treatment for infection, and cases had milder clinical severity.

17. • Lines 293 – 296. Talks about “a guarantee of similar exposure risk” but this is untrue. It’s true that mandatory screening may improve case ascertainment, leading to identification of a high proportion of infections – this is a real strength of the study. But there is no guarantee that vaccinated/unvaccinated people behave in a similar way or arise from the same source populations. For geographical, demographic or other reasons those populations maybe very different. You cannot account for this through adjustment in the study because the data are not available. This is an important limitation of the study which is mentioned, but only briefly. The negative VEs observed in table 2 may be caused by this (if vaccinated people experience more exposure than their unvaccinated peers).

Response: Thanks. We agree with the reviewer. Our statement, “a guarantee of similar exposure risk”, seems arbitrary. We have revised this sentence as following (page 5, line 188-190):

...the difference in exposure risk was partly reduced and population vaccination coverage between intervention groups and control groups were comparable.

Unfortunately, we don’t have geographic features to be controlled or matched, and have added this point to limitation part as follows (page 5, line 202-204):

Some key variables, such as occupational exposure, neighborhood or other spatial feature, living environment (community or nursing home), were not available and may not have been balanced, leading to an unknown bias in an unknown direction.

18. In summary, I think this study has a lot of potential and I commend the author for their work. A simpler and clearer analysis is needed; with more explanation of limitations. I would also like to see recommendations about the frequency of boosting – you have data to describe VE as a function of time since last dose. This can be very important for control of the current Chinese epidemic.

Response: Thanks for your comment. We have made a clearer description of limitations. The frequency of boosting vaccination (3rd-dose) is divided into 181 ~ 270 days, 271 ~ 360 days, and 361 and above. However, the number of cases of severe/critical Covid-19 and Covid-19 related death over a longer time interval was small, and the VE estimates were not sufficiently robust to provide reliable evidence.

REVIEWER COMMENTS

Reviewer #1 (Remarks to the Author):

The authors did not reply or partially reply the majority of this reviewer concerns.

Methods

1 - To emulate a target trial, the authors should mimic inclusion/exclusion criteria and allow for the circumstances to match what would occur in a trial. Therefore, there are two major flaws on it, because two exclusion criteria the authors used before matching.

Problem 1 (Cohort 1 - 2): exclusion of those with follow-up time 0-7 days (flowchart, figure 1). The authors cannot exclude these individuals before matching, this is similar to randomizing someone only after 7 days of randomization, creating a post-randomization bias by conditioning on future. These individuals should be entered in the matching pool (as they would be randomized) and, could not be analysed, but they must enter in the matching pool.

Problem 2 (Cohort 1-2): exclusion of those who did not follow the recommended schedule. This has some ground if the authors want to exclude potential errors in the database, or those immunosuppressed that could follow a different schedule, but this cannot be done for those, for instance, that took the second dose after 59 days. These are natural to occur, represent a population that might be at high risk of infection, etc. They would occur in a RCT too, and to keep an ITT, they would be included in the analysis.

The arguments used in the reply do not hold to support excluding those that did not follow the scheme. The provided VEs in the reply shows this bias.

2 - Both reviewers asked to the authors to provide a clear VE for 1 dose, 2 doses and 3rd those. This is not clear in the manuscript neither in the reply. It seems in Table 3 there are VEs for time following 2 doses and after the third dose. Please, correct it and report it as recommended through the manuscript, instead of " ≥ 1 dose".

3 - Several estimates have wide confidence intervals, showing that precision is a problem. This occurs for the whole population and even worse for subgroups, showing unexpected results. The authors should temper their comments because of this.

3a. Look at VE of 2 doses, ≥ 180 days, above 70, it is $\sim 40\%$ for severe/critical covid. this is a major public health problem and explains very likely what have been shown on the recent outbreak in China.

4 - Reading the criteria the authors copy and paste from the Technical guidelines, it is clear they are hard to follow with administrative data. How could the authors assure to exclude those with fever, acute illness, uncontrolled chronic diseases in the control eligible individuals? This is not feasible.

5 - Regarding the writing, the authors still mention those never vaccinated. This is imprecise, since controls could be vaccinated after matching (the correct way to run the target and rolling country cohort, and the authors did). Please, rephrase.

6 - Please, cite that the source list was not population registry, but those tested.

7 - Replacement. This must be cited in the methods. Furthermore, the authors did not perform any sensitivity analysis, and their arguments about the SE citing other studies is not a support for their own data. The amount of replacement is very high and a sensitivity analysis should be done. Additionally, the potential problem with replacement is not only SE, but the meaning and interpretation, particularly in exact matching.

8 - VoC. The authors must split their analysis, for Cohort 1 and 2, among Omicron and non Omicron period, at minimum. There is no meaning to show a unique VE for Omicron + non

Omicron period.

9 - VE by Brand. The literature clearly shows differences in the immune response and some VE between inactivated vaccines. If the authors have this analysis, there is no reason not to show them, since it has public health interest. Please, add this analysis to the manuscript.

10 - Regarding the VE of the period 0-7 or 0-14 days. This is a "bias indicator", since we did not expect any biological change, any VE in this period is due to residual bias and confounding. The authors found 100% VE in this period, thus, showing very high bias. Please, reconsider the inclusion criteria and use this VE as an indicator of bias, as has been done in several VE studies, including the initial landmark Dagan, NEJM.

11 - The discussion and interpretation is still overoptimistic. The VE of subgroups, the short follow-up, not including Omicron vs non Omicron period, all have a different scenario to be reported and discussed.

12 - Please, verify the KM follow-up, using the start and end date, the maximum fw time is 253 days for the booster dose, but the figure goes beyond 270 days.

Reviewer #2 (Remarks to the Author):

Thank you for your responses. I have some follow-up comments, listed below, focussing mainly on time-dependent risks and additional explanation of control measures given the distribution of cases in time:

Previous comment

There is inadequate context on COVID-19 and related control measures in Shanghai. During the study period I guess SARS-CoV-2 circulation was minimal; I guess lockdowns were standard; and I guess cases were mostly ascertained from mandatory PCR screening. We need also to know what kind of NPIs were in place over the study period and how they may have impacted transmission and the likelihood to vaccinate.

Response: We appreciate the suggestion. We have added following sentences to the introduction (page 2, line 55-60): ...and the citywide vaccination campaign was suspended during the outbreak. The epidemic prevention and control strategy of Covid-19 changed from a targeted approach (March 2021 to March 2022), to strict non-pharmaceutical interventions (NPIs), such as citywide home quarantine, massive nucleic acid amplification testing/rapid antigen testing (NAAT/RAT), and centralized quarantine of close contacts (March 2022 to May 2022), to regular periodic routine NAAT (May 2022 to Dec 7 2022).

New comments

-> "During this time" (line 53). Which time? Please define.

-> Thank you for the additional explanation. Now that I see that almost all of the cases (98%) were captured within a 2-month period (April and May), I think it would be helpful to be really clear about which measures were in place during those months. Was there a citywide lockdown? Were people allowed out of their apartments/residential compounds? What was the frequency of testing? For everyone, or only for case-contacts? This information should be specific and clear; it is vital to understand the realities of case ascertainment in order to interpret the resulting VEs.

Previous comment

We also need to know how the vaccination campaigns were conducted; and any information possible about the reasons for the low VCR in the elderly. Were there any vaccine mandates or inducements? Was there widespread fear of vaccines?

Response: Vaccination campaigns were voluntary and with informed consent. Our previous work (10.3390/vaccines10010091) showed the willingness to vaccinate the older in their family was around 40% in Shanghai. Recommendations from the government, doctors, friends, and relatives

increased the acceptance of vaccination. If the government and doctors promote the efficacy and safety of vaccines through social media and emphasize the importance of universal vaccination, the public acceptance of vaccination increases.

New comment -> Please provide this information in the paper. It is important for the reader to understand the risk of confounding.

Previous comment

It would be helpful to see an epi curve; and a graph of the evolving vaccination coverage in Shanghai; over time, indicating when the study took place.

Response: Thanks for your advice. Following picture displayed an epi curve and vaccination coverage among elderly in Shanghai.

New comments

-> Please provide this figure or some kind of equivalent information in the manuscript so the reader understands the distribution of included cases in calendar time. This is especially important because such a high proportion occurred in such a narrow time window.

Previous comment

The term "to emulate clinical trials" is used several times and should be removed. This is a classical observational study design, all of which would hope to emulate clinical trials, but this study has no more of a claim to draw causal conclusions than any other (in fact due to the minimal control for confounders, it is liable to residual confounding).

Response: We agree with the reviewer that our analysis is liable to confounding caused by other potential factors, although we have controlled age, sex, kinds of chronic diseases, and time since completion of 2-dose vaccine schedule. As you suggested, we have removed the term "emulate clinical trials" in main text except for the Method section, and the abstract.

New comment

-> My recommendation is based on the fact that you did not – as far as I can tell – emulate a clinical trial. For a better understanding of this process, please refer to <https://www.ncbi.nlm.nih.gov/pmc/articles/PMC8010592/> or <https://www.ncbi.nlm.nih.gov/pmc/articles/PMC4832051/>. If you wish to claim you emulated a clinical trial, please explain the process you followed [probably based on an existing clinical trial] which differentiates your study from a classical cohort study in which matching and adjustment are common. Or, remove the claim that you are emulating a clinical trial.

Previous comment

"Adjusted" results are presented but there's no mention of this in the methods. Controlling for matched variables can be problematic and I think should be the subject of a sensitivity analysis.

Response: Thanks. We have re-analyzed following your suggestion, and updated analysis results.

New comment

-> Please explain in the methods section which variables were adjusted for; and the process of model selection.

Previous comment

How was rVE calculated? Not mentioned in the methods

Response: "rVE" in this manuscript referred to VEs of 3-dose vaccination comparing to 2-dose vaccination of inactivated Covid-19 vaccine. We have added following sentence to methods section (page 6, line 237-239): Cohort 1 was used to estimate absolute VEs against Covid-19 outcomes, and cohort 2 estimated relative VEs of booster vaccination by comparing Covid-19 outcomes between the booster vaccination group and the fully (2-dose) vaccinated group

New comment -> please explain in the methods how rVE was calculated (perhaps the formula

was: 1-HR of 3d vs 2d group?). "comparing Covid-10 outcomes", which you have written, isn't really clear from a statistical perspective.

Previous comment

Lines 293 – 296. Talks about "a guarantee of similar exposure risk" but this is untrue. It's true that mandatory screening may improve case ascertainment, leading to identification of a high proportion of infections – this is a real strength of the study. But there is no guarantee that vaccinated/unvaccinated people behave in a similar way or arise from the same source populations. For geographical, demographic or other reasons those populations maybe very different. You cannot account for this through adjustment in the study because the data are not available. This is an important limitation of the study which is mentioned, but only briefly. The negative VEs observed in table 2 may be caused by this (if vaccinated people experience more exposure than their unvaccinated peers).

Response: Thanks. We agree with the reviewer. Our statement, "a guarantee of similar exposure risk", seems arbitrary. We have revised this sentence as following (page 5, line 188-190): ...the difference in exposure risk was partly reduced and population vaccination coverage between intervention groups and control groups were comparable.

Unfortunately, we don't have geographic features to be controlled or matched, and have added this point to limitation part as follows (page 5, line 202-204): Some key variables, such as occupational exposure, neighborhood or other spatial feature, living environment (community or nursing home), were not available and may not have been balanced, leading to an unknown bias in an unknown direction.

New comment -> You say "...and population vaccination coverage between intervention groups and control groups were comparable." How do you know this is true? As far as I can tell you have no information on the VCR in the community and the opposite of this statement may be true.

New comment - table 4. Please change "VE" to "rVE".

Response to Reviewers

We thank the Editor and the two anonymous reviewers for their constructive inputs to our paper. We have addressed all the comments. Please find our point-to-point responses below.

Reviewer # 1 (Remarks to the Author):

The authors did not reply or partially reply the majority of this reviewer concerns.

Methods

1. To emulate a target trial, the authors should mimic inclusion/exclusion criteria and allow for the circumstances to match what would occur in a trial. Therefore, there are two major flaws on it, because two exclusion criteria the authors used before matching.

Problem 1 (Cohort 1 - 2): exclusion of those with follow-up time 0-7 days (flowchart, figure 1). The authors cannot exclude these individuals before matching, this is similar to randomizing someone only after 7 days of randomization, creating a post-randomization bias by conditioning on future. These individuals should be entered in the matching pool (as they would be randomized) and, could not be analysed, but they must enter in the matching pool.

Response: We agree with the reviewer that individuals with follow-up time 0-7 days should enter the matching pool, and we have done so as suggested. In addition, we have excluded such individuals and their matched controls from the analysis.

Problem 2 (Cohort 1-2): exclusion of those who did not follow the recommended schedule. This has some ground if the authors want to exclude potential errors in the database, or those immunosuppressed that could follow a different schedule, but this cannot be done for those, for instance, that took the second dose after 56 days. These are natural to occur, represent a population that might be at high risk of infection, etc. They would occur in a RCT too, and to keep an ITT, they would be included in the analysis.

The arguments used in the reply do not hold to support excluding those that did not follow the scheme. The provided VEs in the reply shows this bias.

Response: Thanks for this comment. We have re-analyzed the data following your suggestion in which we included the individuals taking the second dose after 56 days, and updated analysis results (table 2-4).

2. Both reviewers asked to the authors to provide a clear VE for 1 dose, 2 doses and 3rd those. This is not clear in the manuscript neither in the reply. It seems in Table 3 there are VEs for time following 2 doses and after the third dose. Please, correct it and report it as recommended through the manuscript, instead of "≥1 dose"

Response: Thanks. We have provided VEs for those who took 1 dose, 2 doses and 3 dose respectively (table 3).

3. Several estimates have wide confidence intervals, showing that precision is a problem. This occurs for the whole population and even worse for subgroups, showing unexpected results. The authors should temper their comments because of this.

3a. Look at VE of 2 doses, 180 days, above 70, it is ~40% for severe/critical covid. This is a major public health problem and explains very likely what have been shown on the recent outbreak in China

Response: Thanks a lot for your comment. We agree with your point that the VE of inactivated vaccines seems to decay quickly and may lead to the high proportion of infection in the recent outbreak in China. We have revised the discussion and interpretation accordingly.

4. Reading the criteria the authors copy and paste from the technical guidelines, it is clear they are hard to follow with administrative data. How could the authors assure to exclude those with fever, acute illness, uncontrolled chronic diseases in the control eligible individuals? This is not feasible.

Response: Thanks a lot for your comment. In China Covid-19 vaccination campaign is under the principle of informed consent. Before receiving Covid-19 vaccines, each people need to spontaneously report their current health conditions to medical staffs who would evaluate whether they were eligible for vaccination.

5. Regarding the writing, the authors still mention those never vaccinated. This is imprecise, since controls could be vaccinated after matching (the correct way to run the target and rolling country cohort, and the authors did). Please, rephrase.

Response. Thanks. We didn't find the item "never vaccinated" in the paper. As the reviewer said the controls in our study could be vaccinated after matching. If the controls vaccinated, they and their matched vaccinated individuals were censored from the analysis.

6. Please, cite that the source list was not population registry, but those tested.

Response: Thanks. We have explained the source list in method section as following (lines 224-225):

... The source population were derived from individuals undergoing several rounds of citywide massive NAAT/RAT, which involved everyone living in Shanghai including citizens, foreigners, and immigrants...

7. Replacement. This must be cited in the methods. Furthermore, the authors did not perform any sensitivity analysis, and their arguments about the SE citing other studies is not a support for their own data. The amount of replacement is very high and a sensitivity analysis should be done. Additionally, the potential problem with replacement is not only SE, but the meaning and interpretation, particularly in exact matching.

Response: Thanks. As suggested, we have performed a sensitivity analysis using

sampling without replacement and displayed the results in paper (table S2).

8. VoC. The authors must split their analysis, for Cohort 1 and 2, among Omicron and not Omicron period, at minimum. There is no meaning to show a unique VE for Omicron + non Omicron period.

Response: Thanks. We have added a new picture in this paper (figure 1), and as you can see nearly 96% of documented cases in our analysis occurred between March 2022 and May 2022, which assumed to be the omicron dominant period (10.1016/S0140-6736(22)00838-8).

9. VE by Brand. The literature clearly shows differences in the immune response and some VE between inactivated vaccines. If the authors have this analysis, there is no reason not to show them, since it has public health interest. Please, add this analysis to the manuscript.

Response: Thanks, and we have added VEs by brand in the paper (table S3).

10. Regarding the VE of the period 0-7 or 0-14 days. This is a "bias indicator", since we did not expect any biological change, any VE in this period is due to residual bias and confounding. The authors found 100% VE in this period, thus, showing very high bias.

Please, reconsider the inclusion criteria and use this VE as an indicator of bias, as has been done in several VE studies, including the initial landmark Dagan, NEJM.

Response: Thanks. We have provided the VEs of the periods 0-7/0-14 days in manuscript (table S3).

11. The discussion and interpretation are still overoptimistic. The VE of subgroups, the short follow-up, not including Omicron vs non Omicron period, all have a different scenario to be reported and discussed.

Response: Thanks. The longest follow-up time in cohort 1 was 478 days (median 282), and in cohort 2 it was 274 (median 76), which are actually longer than previously published literatures (10.1016/S1473-3099(22)00292-4, 10.1056/NEJMoa2200797, and 10.1016/j.lanep.2022.100466). We have revised the discussion and interpretation.

12. Please, verify the KM follow-up, using the start and end date, the maximum fw time is 253 days for the booster dose, but the figure goes beyond 270 days.

Response: Thanks for your comment, and we have verified the KM follow-up using the start and end date.

Reviewer # 2 (Remarks to the Author):

Thank you for your responses. I have some follow-up comments, listed below, focusing mainly on time-dependent risks and additional explanation of control measures given the distribution of cases in time:

1. Previous comment

There is inadequate context on COVID-19 and related control measures in Shanghai. During the study period I guess SARS-CoV-2 circulation was minimal; I guess lockdowns were standard; and I guess cases were mostly ascertained from mandatory PCR screening. We need also to know what kind of NPIs were in place over the study period and how they may have impacted transmission and the likelihood to vaccinate.

Response: We appreciate the suggestion. We have added following sentences to the introduction (page 2, line 55-60): ...and the citywide vaccination campaign was suspended during the outbreak. The epidemic prevention and control strategy of Covid-19 changed from a targeted approach (March 2021 to March 2022), to strict non-pharmaceutical interventions (NPIs), such as citywide home quarantine, massive nucleic acid amplification testing/rapid antigen testing (NAAT/RAT), and centralized quarantine of close contacts (March 2022 to May 2022), to regular periodic routine NAAT (May 2022 to Dec 7 2022).

New comments

-> "During this time" (line 53). Which time? Please define.

-> Thank you for the additional explanation. Now that I see that almost all of the cases (98%) were captured within a 2-month period (April and May), I think it would be helpful to be really clear about which measures were in place during those months. Was there a citywide lockdown? Were people allowed out of their apartments/residential compounds? What was the frequency of testing? For everyone, or only for case-contacts? This information should be specific and clear; it is vital to understand the realities of case ascertainment in order to interpret the resulting VEs.

New Response: Thanks a lot for your comment.

"During this time" referred to "Between March 2021 and May 2022", and we have added it to the manuscript (line 55).

Between March 2021 and May 2022, Shanghai experienced from phased lockdown to strict citywide lockdown. People were not allowed out of their apartments/residential compounds except for NAAT, and basic living supplies (like rice, noodles, vegetables) were provided by the community. Everyone living in Shanghai—citizens, foreigners, and immigrants—underwent several rounds of citywide massive SARS-CoV-2 RT-PCR testing and rapid antigen testing (RAT). Since from 12 April 2022, Shanghai was divided as lockdown zones (neighborhoods that have reported new infections in the last seven days, and residents are required to stay at home for a week under closed-loop management), controlled zones (neighborhoods where no infections were reported in last seven days. Residents are allowed to retrieve food deliveries or take a walk at

designated spots at staggered hours within the compound), and precautionary zones (communities that have not reported infections over the last 14 days. Residents can leave their neighborhood but must stay within their subdistrict, and they are encouraged to limit their movement. Those living in precautionary zones can now move around their neighborhoods, but must observe social distancing and could be sealed off again if there are new infections). The frequency of testing was adjusted by the epidemic severity. Before the division of the three zones, NAAT/RAT was performed every 24 hours. People in lockdown zones received NAAT every 24 hours, in controlled zones received NAAT and RAT alternatively, and in precautionary zones received RAT twice a day.

2. Previous comment

We also need to know how the vaccination campaigns were conducted; and any information possible about the reasons for the low VCR in the elderly. Were there any vaccine mandates or inducements? Was there widespread fear of vaccines?

Response: Vaccination campaigns were voluntary and with informed consent. Our previous work (10.3390/vaccines10010091) showed the willingness to vaccinate the older in their family was around 40% in Shanghai. Recommendations from the government, doctors, friends, and relatives increased the acceptance of vaccination. If the government and doctors promote the efficacy and safety of vaccines through social media and emphasize the importance of universal vaccination, the public acceptance of vaccination increases.

New comment-> Please provide this information in the paper. It is important for the reader to understand the risk of confounding.

New Response: Thanks. We have added this information in the manuscript (line 51-52).

3. Previous comment

It would be helpful to see an epi curve; and a graph of the evolving vaccination coverage in Shanghai; over time, indicating when the study took place.

Response: Thanks for your advice. Following picture displayed an epi curve and vaccination coverage among elderly in Shanghai.

New comments

-> Please provide this figure or some kind of equivalent information in the manuscript so the reader understands the distribution of included cases in calendar time. This is especially important because such a high proportion occurred in such a narrow time window.

New Response: Thanks. We have added this figure in the manuscript (fig. 1).

4. Previous comment

The term "to emulate clinical trials" is used several times and should be removed. This is a classical observational study design, all of which would hope to emulate clinical trials, but this study has no more of a claim to draw causal conclusions than any other (in fact due to the minimal control for confounders, it is liable to residual confounding).

Response: We agree with the reviewer that our analysis is liable to confounding caused by other potential factors, although we have controlled age, sex, kinds of chronic diseases, and time since completion of 2-dose vaccine schedule. As you suggested, we have removed the term "emulate clinical trials" in main text except for the Method section, and the abstract.

New comment

-> My recommendation is based on the fact that you did not - as far as I can tell - emulate a clinical trial. For a better understanding of this process, please refer to <https://www.ncbi.nlm.nih.gov/pmc/articles/PMC8010592/> or <https://www.ncbi.nlm.nih.gov/pmc/articles/PMC4832051/>. If you wish to claim you emulated a clinical trial, please explain the process you followed [probably based on an existing clinical trial] which differentiates your study from a classical cohort study in which matching and adjustment are common. Or, remove the claim that you are emulating a clinical trial.

New Response: As suggested, we have removed the term "to emulate clinical trials" in the paper.

5. Previous comment

"Adjusted" results are presented but there's no mention of this in the methods. Controlling for matched variables can be problematic and I think should be the subject of a sensitivity analysis.

Response: Thanks. We have re-analyzed following your suggestion, and updated analysis results.

New comment

-> Please explain in the methods section which variables were adjusted for; and the process of model selection.

New Response: Thanks. We have explained the selection principle of variables adjusted in model in the methods section (line 308-309) as following:

... *adjusted for comorbidities of hypertension (binary variable, yes or no), type 2 diabetes (binary variable, yes or no), and cancer (binary variable, yes or no) ...*

6. Previous comment

How was rVE calculated? Not mentioned in the methods

Response: "rVE" in this manuscript referred to VEs of 3-dose vaccination comparing to

2- dose vaccination of inactivated Covid-19 vaccine. We have added following sentence to methods section (page 6, line 237-239): Cohort 1 was used to estimate absolute VEs against Covid-19 outcomes, and cohort 2 estimated relative VEs of booster vaccination by comparing Covid-19 outcomes between the booster vaccination group and the fully (2- dose) vaccinated group

New comment-> please explain in the methods how rVE was calculated (perhaps the formula was: 1-HR of 3d vs 2d group?). "comparing Covid-10 outcomes", which you have written, isn't really clear from a statistical perspective.

New Response: Thanks. We explained rVE calculation in methods section (line 310) as following:

... and relative VE was calculated as 1-HR (3-dose group vs 2 dose group in cohort 2).

... "comparing Covid-10 outcomes" was revised as "comparing risks of Covid-10 outcomes" (line 245).

7. Previous comment

Lines 293 - 296. Talks about "a guarantee of similar exposure risk" but this is untrue. It's true that mandatory screening may improve case ascertainment, leading to identification of a high proportion of infections - this is a real strength of the study. But there is no guarantee that vaccinated/unvaccinated people behave in a similar way or arise from the same source populations. For geographical, demographic or other reasons those populations maybe very different. You cannot account for this through adjustment in the study because the data are not available. This is an important limitation of the study which is mentioned, but only briefly. The negative VEs observed in table 2 may be caused by this (if vaccinated people experience more exposure than their unvaccinated peers).

Response: Thanks. We agree with the reviewer. Our statement, "a guarantee of similar exposure risk", seems arbitrary. We have revised this sentence as following (page 5, line 188-190): ...the difference in exposure risk was partly reduced and population vaccination coverage between intervention groups and control groups were comparable.

Unfortunately, we don't have geographic features to be controlled or matched, and have added this point to limitation part as follows (page 5, line 202-204): Some key variables, such as occupational exposure, neighborhood or other spatial feature, living environment (community or nursing home), were not available and may not have been balanced, leading to an unknown bias in an unknown direction.

New comment-> You say "...and population vaccination coverage between intervention groups and control groups were comparable ." How do you know this is true? As far as I can tell you have no information on the VCR in the community and the opposite of this statement may be true.

New Response: Thanks for your comment. Although the daily matching process partly

reduced the difference in exposure risk, without matching of geographic features made population vaccination coverage and VCR across different groups unclear. We have removed this sentence in the paper.

REVIEWER COMMENTS

Reviewer #1 (Remarks to the Author):

This reviewer would like to thank the authors for this revised manuscript and replying to my concerns.

However, with the new results on Table 3, it is clear there is no minimum power to analyse 1 dose, and that the vast majority of VE weighting on the $VE \geq 1$ dose is for, actually, 3 doses. This makes the overall VE presented in Abstract and Table 2 misleading. Overall, I would say the authors are in place for relative VE only in terms of power and reliability, which has a lot of value per se.

This reviewer strongly suggests to the authors to

- 1) Run an analysis of VE at least 1, at least 2, at least 3 doses as the main analysis (not applying subgroups neither time stratification).
- 2) Delete Table 3, because it has no meaning in it

It is hard to follow the storytelling of the manuscript, and as stated since the first review, there is no meaning to Public Health and individuals decision saying $VE \geq 1$ dose at this time of the pandemic. Indeed, the correct message supported by data could say that not only vaccine uptaking, but completing the booster calendar, which is not a message send by $VE \geq 1$ dose.

Reviewer #2 (Remarks to the Author):

My comments have been addressed - thank you.

Response to Reviewers

We thank the Editor and the two anonymous reviewers for their constructive inputs to our paper. We have addressed all the comments. Please find our point-to-point responses below.

Reviewer #1 (Remarks to the Author):

This reviewer would like to thank the authors for this revised manuscript and replying to my concerns. However, with the new results on Table 3, it is clear there is no minimum power to analyse 1 dose, and that the vast majority of VE weighting on the $VE \geq 1$ dose is for, actually, 3 doses. This makes the overall VE presented in Abstract and Table 2 misleading. Overall, I would say the authors are in place for relative VE only in terms of power and reliability, which has a lot of value per se.

Response: We agree with this general comment.

1) Run an analysis of VE at least 1, at least 2, at least 3 doses as the main analysis (not applying subgroups neither time stratification).

Response: Thanks a lot for your comment. Analysis of VEs at least 1 dose have displayed in table 2, and added analysis of VEs at least 2 and at least 3 doses have put in revised table 3.

2) Delete Table 3, because it has no meaning in it

Response: Done as suggested.

It is hard to follow the storytelling of the manuscript, and as stated since the first review, there is no meaning to Public Health and individuals decision saying $VE \geq 1$ dose at this time of the pandemic. Indeed, the correct message supported by data could say that not only vaccine uptaking, but completing the booster calendar, which is not a message send by $VE \geq 1$ dose.

Response: Thanks for arising this important point. We also revised the abstract and conclusions on the importance of the booster dose accordingly.

Reviewer #2 (Remarks to the Author):

My comments have been addressed - thank you.

Response: Thanks a lot for your constructive inputs to our paper.